# RECURRENT NEURAL CELLULAR AUTOMATA WITH SELF-ATTENTION FOR MULTI-AGENT SYSTEM

## ABSTRACT

Multi-agent systems, such as epidemic spread, rumor propagation through crowd, prey-predator model, and forest fire, exhibit complex global dynamics originated from local, relatively simple, and often stochastic interactions between agents. Despite significant advancements in predictive modeling through deep learning, such interactions among many agents have rarely explored as a specific domain for predictive modeling. We present **R**ecurrent **A**ttention-based **N**eural **C**ellular **A**utomata (RA-NCA), to effectively discover the local stochastic interaction by associating the temporal information between neighboring agents in a permutation-invariant manner. RA-NCA exhibits the superior generalizability across various agent configurations (i.e., spatial distribution of agents), data efficiency and robustness in extremely data-limited scenarios even with the presence of stochastic interactions, and scalability through spatial dimension-independent prediction. We compare and evaluate RA-NCA with other NCA networks and scene prediction networks in the three synthetic multi-agent systems with thousands of agents, such as forest fire, host-pathogen, and stock market models.

## 1 INTRODUCTION

In natural systems, seemingly simple local interactions between agents often give rise to intricate behaviors. Consider, for instance, the spread of epidemics (Ghosh & Bhattacharya, 2020; White et al., 2007), the propagation of rumors in social networks (Wang et al., 2014; Kawachi et al., 2008), the dynamics governing forest fires (Karafyllidis & Thanailakis, 1997; Trunfio, 2004), and the swarm of birds, fish, and insects (Couzin et al., 2003; Ha & Tang, 2022). These systems with massive amount of agents (1000 $\sim$) are intrinsically linked to our daily lives. The ability to predict their future states a valuable asset for informed decision-making and long-term planning. Despite substantial advancements in predictive modeling facilitated by deep learning techniques, locally-interacting multi-agent systems have remained relatively underexplored as specific prediction problems. One notable distinction in such systems lies in that slightly modified configuration of agents, such as their spatial distribution or non-deterministic interaction, can lead to significantly different future states, which is a challenging generalization problem in many deep learning models.

Traditional cellular automata with human-designed local interaction rules have long served as an effective computational model for simulating above-mentioned multi-agent systems (Bauer et al., 2009; Macal & North, 2009; Clarke, 2014). However, in the real world, we often lack complete or even partial knowledge of the interaction rules governing these systems (Banks & Hooten, 2021). Recent progress in cellular automata has embraced deep learning, which is often referred to as neural cellular automata (NCA) (Gilpin, 2019; Grattarola et al., 2021; Tesfaldet et al., 2022; Mordvintsev et al., 2020; Sudhakaran et al., 2021). NCA can design the interaction rules in a data-driven approach using neural networks without tricky assumptions or knowledge about the systems. Previous works typically utilized simple and shallow convolutional neural network (CNN)-based architectures with a $3 \times 3$ receptive field to capture information from neighboring cells effectively. Nonetheless, these applications have predominantly focused on static and deterministic environments such as image processing, where historical information about current cells is not necessarily required to predict their future states. The application of NCA to multi-agent systems driven by dynamic and stochastic local interactions has been unexplored, and the design of NCA networks for such application is also missing.

In this paper, we propose **R**ecurrent **A**ttention-based **N**eural **C**ellular **A**utomata (RA-NCA), which learns the local interaction using the temporal information of agents in a permutation-invariant manner. RA-NCA involves a novel recurrent cellular attention module that couples long short-term memory (LSTM) (Hochreiter & Schmidhuber, 1997) and cellular self-attention. Cellular self-attention module associates the hidden states of each of agents in Moore's neighborhood and models the next hidden states together with LSTM. In particular, it removes the spatial dependency in neighboring agents, hence the interaction between the center agent and neighboring agents from any directions can effectively contribute to the learning process. RA-NCA is evaluated in the task to predict the probability of agents to be on a certain state within the three synthetic locally-interacting multi-agent systems; forest fire model (Tisue & Wilensky, 2004), host-pathogen model (Sayama, 2013), and stock market model (Wei et al., 2003). RA-NCA exhibits superior performance than other NCA networks and scene prediction networks in all the three datasets with various stochasticity levels.

Our technical novelty is primarily in identifying the interesting properties emerged from this novel NCA-based architecture in locally interacting multi-agent systems. We empirically demonstrate the three advantages of RA-NCA. The permutation invariance in RA-NCA leads to efficient discovery of the hidden interaction rules in extremely data-limited scenarios (**Data efficiency**). We observe that such data efficient learning is even consistent in the presence of stochastic interactions (**Robustness**). Furthermore, RA-NCA is flexible in the prediction dimension due to the agent-based processing. We show that RA-NCA trained in relatively small systems can be successfully applied to $64\times$ larger systems without re-training and degradation of the performance (**Scalability**).

## 2 RELATED WORK AND BACKGROUND

**Neural Cellular Automata** Neural Cellular Automata (NCA) have been studied in various non-dynamic tasks (dealing with stationary data), such as image denoising (Tesfaldet et al., 2022) and texture generation (Pajouheshgar et al., 2023; Mordvintsev et al., 2020). Recently, other interesting applications to discrete systems have been also explored, including damage recovery of soft robots (Horibe et al., 2021) and various game map generation (Earle et al., 2022). The prior methods are mostly built on CNN-based architectures and replace conventional CA processes into iterative inferences in neural networks. However, they are potentially challenging to learn dynamical systems due to the following two reasons. First, the NCAs without memory functions hinders modeling the dynamic cells if the transition rule of cells is depending on their temporal information. Second, the iterative inferences for sequential data (not a single image) create a long computational graph, which makes both training and inference unstable and computationally expensive. Further discussion and empirical results on state-of-the-art NCA network (Tesfaldet et al., 2022) are available in Appendix C. In this light, a novel NCA architecture, such as recurrent neural networks (RNN), is required to expand the benefits of NCAs to the applications in dynamical systems.

**Interaction Learning in Multi-agent System** Prior works in multi-agent interaction learning mostly address few number of agents ($\sim 10$) using a complete graph, assuming that all agents are interacting each other (Sankararaman et al., 2019; Liu et al., 2020; Saha et al., 2020; Li et al., 2021; Niu et al., 2021; Sun et al., 2023), or pre-defined interaction graphs (Battaglia et al., 2016; Huang et al., 2020). However, our systems are structured with massive number of locally interacting agents (1000 $\sim$). Complex physical systems with many nodes have been also explored using graph networks (Zang & Wang, 2020; Sanchez-Gonzalez et al., 2020). Their methods can address the locally interacting systems, but mostly aim to simulate the trajectory of particles, where agent states are not explicitly defined. In contrast, our focus lies in modeling the transition of agent states. Also, RA-NCA exhibits low computational complexity (linear to the number of agents) due to the CA-based local processing, while the computational cost in other methods often quadractically increases (Zang & Wang, 2020). Reinforcement learning (RL) has been also considered in locally interacting multi-agent systems (Zhang et al., 2018), but generally focusing on control problems, not prediction problems. Relatively few have investigated modeling sequential data (Janner et al., 2021; Ganapathi Subramanian & Crowley, 2018). However, RL-based approaches generally exhibit low sample efficiency (Micheli et al., 2022), which could be critical to address our extremely data-limited scenarios.

**Dataset: Synthetic Multi-agent System** Multi-agent systems in this paper specifically refers a system where stationary pixel agents with discrete states are locally interacting with their neighboring agents. The agent interaction is defined by the pre-defined rules that change the state of agents. We use the three synthetic multi-agent systems, forest fire, host-pathogen, and stock market models, as our training and evaluation environments. There are four agent states in each of the systems, as described in Figure 1. For example, the host-pathogen model involves empty, dead, healthy, and infected agents. The host-pathogen model evolves following the sequence of interaction rules: **1)** Create 64×64 agent map filled with randomly distributed infected (1%), healthy (75%), and empty (the rest) agents. **2)** For dead agents, cured by each of neighboring healthy agents with the probability $p_{\text{cure}}$. **4)** For healthy agents, infected by each of neighboring infected agents with probability $p_{\text{infect}}$. **5)** For infected agents, transitioned to dead agents. Repeat from **2)** to **5)**. The forest fire and stock market models also have their own interaction rules. In the forest fire model, fire agents (randomly ignited at initialization) radiate heat to neighboring tree agents with the probability $p_{\text{heat}}$ (deterministic if $p_{\text{heat}} = 1$), and if the accumulated heat on trees exceeds a threshold, tree states are transitioned to fire states. In the stock market model, there is a transition matrix parameterized by $p_{\text{invest}}$ to model the stochastic transition between investor's actions (buy, hold, sell, and inactive). The dominant action of neighboring agents determines which element in the transition matrix the center agent will follow. More details are available in Appendix A. While this paper primarily investigates the performance of RA-NCA in the synthetic environments, RA-NCA also exhibits promising results in the real-world application such as wildfire prediction. (see Appendix E)

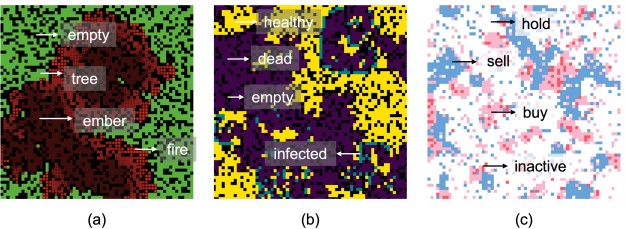

(a)          (b)          (c)

Figure 1: Synthetic locally-interacting multi-agent systems (a) forest fire model, (b) host-pathogen model, (c) stock market model. All the systems are defined on 64×64 agent maps, and their states are represented by distinct colors.

## 3 Proposed Approach

We propose a novel NCA architecture (RA-NCA) specifically designed for discovering unknown local interaction rules in multi-agent dynamic systems. Given the recent successes in NCAs, the detailed motivation of the need for novel NCA architectures for such systems is discussed.

**Lack of memory function in existing NCA** Most prior NCA networks are based on CNN (Mordvintsev et al., 2020; Pajouheshgar et al., 2023; Gilpin, 2019). Recently, graph-based (Grattarola et al., 2021), attention-based (Tesfaldet et al., 2022), variational autoencoder-based NCA (Palm et al., 2022) networks are also proposed. However, RNN has been rarely coupled with NCA. Memory is an important function to model the evolution of multi-agent systems because the next state or action of agents can often be a function of the past states. For example, in forest fire model, when agents on fire states are transitioned to ember states, so called new ember states, it can still radiate heat to neighbor agents while old ember states do not. In this case, the time when the fire state is transitioned to the ember state is important information to accurately model the heat accumulation in neighbor agents. Also, consider the spread of epidemics. Infected agents do not immediately pass on other agents. It requires a certain latent period for infected agents to be infectious.

**Spatial dependency in CNN-based NCA** While CNN is a prevalent form of NCA networks, it may not be effective in the multi-agent systems. The fundamental bottleneck in CNN is the spatial dependency when it estimates interaction between agents. Figure 2 shows the two different configuration of agents in 3×3 cells. Let us assume that the red agent is interacting with the blue, and white is inactive. Figure 2(a) describes convolution on the red agent as an inner product between

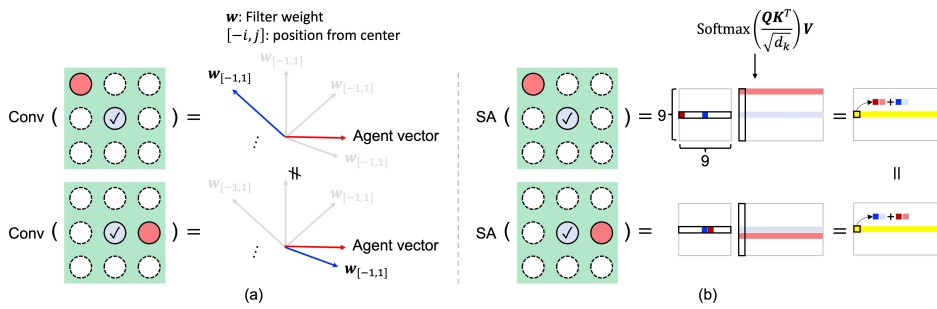

Figure 2: Spatial dependency on agent location in (a) convolution and (b) self-attention.

the weight vector (blue) and the encoded vector of the agent (red). However, the weight vector is not identical across the different agent locations, and the convolution output depends on the spatial order of neighboring agents. That is, the nine weight vectors should be properly shifted together to appropriately measure the interaction regardless of the location of the red agent. This requires the model to be exposed by a various set of spatial configurations, possibly by more training samples, so that all the weight vectors can be updated. In contrast, Figure 2(b) shows that self-attention models the interaction through attention scores, which is represented as a 9×9 matrix, and the location of the red agent is no longer important in the final yellow vector. Consequently, the training of self-attention based networks independent to the spatial order of the red agents and directly learning the interaction. Our experimental results support the biased and balanced interaction learning in CNN-based approach and attention-based approach. More details are available in Appendix B.

## 3.1 RECURRENT ATTENTION-BASED NEURAL CELLULAR AUTOMATA NETWORK

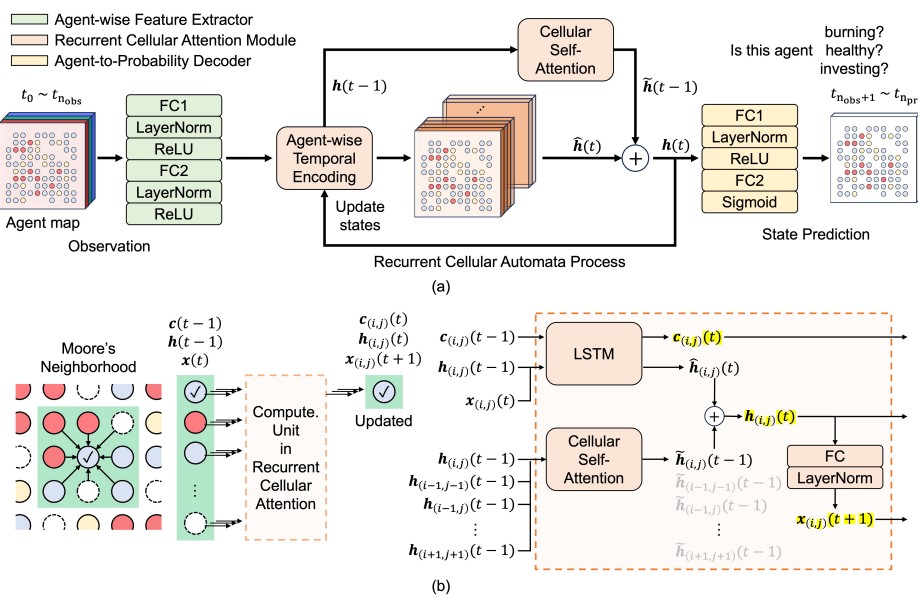

Figure 3: Recurrent Attention-based Neural Cellular Automata network.

Cellular automata is generally designed with the cell states and update rules as a function of the states (Mordvintsev et al., 2020). We aim to design a neural network-based non-trivial function (i.e., update rules) operating on the continuous vectors that represent the states of agents (i.e., cell states). Mathematically, our design space is mainly defined on $x_{(i,j)}(t+1) = f(x_{(i,j)}(t), x_{S_{(i,j)}(1)}, ..., x_{S_{(i,j)}(8)}(t)) : \mathbb{R}^n \to \mathbb{R}^n$, where $x_{(i,j)}(t)$ is the state vector of an agent at the location $(i,j)$ and time $t$, and $S_{(i,j)}$ is a set of eight neighboring agent's locations from $(i,j)$.

It should be permutation invariant as following:

$$f(x_{(i,j)}(t), x_{S_{(i,j)}(1)}(t), x_{S_{(i,j)}(2)}(t), ..., x_{S_{(i,j)}(8)}(t)) =$$
$$f(x_{(i,j)}(t), x_{S_{(i,j)}(\pi(1))}(t), x_{S_{(i,j)}(\pi(2))}(t), ..., x_{S_{(i,j)}[\pi(8)]}(t)) \tag{1}$$

where $\pi$ is every permutation of $\{1, 2, ..., 8\}$. In addition to $f$, we need two more functions. The agents in our systems are assumed to be observed in RGB space. Also, the ultimate task is to make the probabilistic estimation of an agent's certain state in the future. Hence, we need an encoding function that transfers the RGB vectors into high-dimensional vectors $\in \mathbb{R}^n$ and a decoding function that converts the high-dimensional vectors into probabilities. In summary, RA-NCA is composed of the three functions; agent-wise feature extractor ($\mathbb{R}^3 \rightarrow \mathbb{R}^n$), recurrent cellular attention module ($f : \mathbb{R}^n \rightarrow \mathbb{R}^n$), and agent-to-probability decoder ($\mathbb{R}^n \rightarrow \mathbb{R}^1$).

Figure 3(a) shows the overview of the proposed network architecture. RA-NCA observes $t_{n_{obs}}$ successive scenes (i.e., agent maps) of the system and predicts the next $t_{n_{pred}} - t_{n_{obs}}$ scenes as probability maps. The agent-wise feature extractor processes the individual agent's state represented by RGB colors during the observation period. The recurrent cellular attention module encodes the observed sequence of individual agent's states (agent-wise temporal encoding), associates the encoded temporal information with neighboring agents by cellular self-attention, and predicts the next sequence in an autoregressive manner. The agent-to-probability decoder converts the encoded and predicted agent's states into probability values and creates probability maps.

The feature extractor consists of two fully-connected layers ($\mathbb{R}^3 \rightarrow \mathbb{R}^{32}$). It is shared across entire agents since there is no spatial correlation in agent states. Note, the spatial dimensions of an agent map are not changed in the encoded space, and only the channel dimension varies. It is important to preserve the spatial dimensions because the interaction is very localized, indicating the individual agent's behavior is critical in modeling the interaction rules. The decoder is also designed with two fully-connected layers and shared for all the agents ($\mathbb{R}^{32} \rightarrow \mathbb{R}^1$). As the final output should be a probability, sigmoid function is used as the last activation. Ground truth for the prediction is given by binary maps. The network is trained by binary cross-entropy (BCE) loss:

$$\mathcal{L} = -\frac{1}{N(t_{n_{pred}} - 1)} \sum_{t=1}^{t_{n_{pred}}-1} \sum_{i,j} y_{(i,j)}(t) \log \hat{y}_{(i,j)}(t) + (1 - y_{(i,j)}(t)) \log (1 - \hat{y}_{(i,j)}(t)) \tag{2}$$

where $y$ and $\hat{y}$ are the binary ground truth for an agent's certain state and predicted probability of the state, respectively, and $N$ is the total number of agents. Training hyperparameters are noted in the experiment setting section.

Our key contribution is in the recurrent cellular attention module, which is the distinct module of RA-NCA. The motivation is that, the dynamics of each agent is updated to mimic the dynamics of neighboring agents in the locally-interacting multi-agent systems. Considering the propagation of forest fire, spread of epidemics or rumors, the dynamics of each agent is totally altered after the interaction, and the new dynamics is analogous to the dynamics of neighbor that induced the interaction. In this light, recurrent cellular attention module aims to efficiently deliver the dynamics of neighboring agents into the center agent.

In order to deal with the agent-wise dynamics, we assume that the temporal information of each agent should be well-preserved in the module. For the reason, the agent-wise temporal encoding is performed by a single-layer long short-term memory (LSTM) which is shared across the entire agents while having the independent hidden states for each of agents (Hochreiter & Schmidhuber, 1997). Note, RA-NCA with a basic RNN module (not LSTM) is investigated in Appendix B. Now, the dynamics of each agent is translated into its hidden states in LSTM. Cellular self-attention aims to quickly update the hidden states of each agent by delivering the dynamics of neighboring agents. Figure 3(b) describes the recurrent cellular attention mechanism in Moore's neighborhood. Formally, the hidden state ($\mathbf{h}_{(i,j)}(t)$) for an agent at $(i, j)$ position at time $t$ in recurrent cellular attention is given by:

$$\mathbf{h}_{(i,j)}(t) = \tilde{\mathbf{h}}_{(i,j)}(t-1) + \hat{\mathbf{h}}_{(i,j)}(t) \tag{3}$$

where $\tilde{\mathbf{h}}_{(i,j)}(t-1)$ is the output from the cellular self-attention module, representing the attention vector for the agent at $(i, j)$ associated with its' Moore's neighborhood, and $\hat{\mathbf{h}}_{(i,j)}(t)$ is the hidden

state modeled from the LSTM cell for the agent at $(i, j)$ at time $t$. An intuitive understanding for (3) is that, by summing the $(i, j)$ agent's hidden state and its neighbor's one, the dynamics of $(i, j)$ agent is quickly shifted to mimic the behavior of its neighbor. $\tilde{\mathbf{h}}_{(i,j)}(t-1)$ and $\hat{\mathbf{h}}_{(i,j)}(t)$ are given by:

$$\hat{\mathbf{h}}_{(i,j)}(t) = \mathbf{o}_{(i,j)}(t) \odot \tanh(\mathbf{c}_{(i,j)}(t)) \tag{4}$$

$$\tilde{\mathbf{h}}_{(i,j)}(t-1) = f_\theta(\text{softmax}(\mathbf{Q}_{S_{(i,j)}} \mathbf{K}_{S_{(i,j)}}^{\mathsf{T}}) \mathbf{V}_{S_{(i,j)}})_{(i,j)} \tag{5}$$

$$\mathbf{Q}_{S_{(i,j)}} = \mathbf{W}_{\mathbf{Q}} \mathbf{h}_{S_{(i,j)}}(t-1); \ \mathbf{K}_{S_{(i,j)}} = \mathbf{W}_{\mathbf{K}} \mathbf{h}_{S_{(i,j)}}(t-1); \ \mathbf{V}_{S_{(i,j)}} = \mathbf{W}_{\mathbf{V}} \mathbf{h}_{S_{(i,j)}}(t-1) \tag{6}$$

where $S_{(i,j)}$ is a set of agents in Moore's neighborhood of the agent at $(i, j)$, $\mathbf{c}_{(i,j)}$ and $\mathbf{o}_{(i,j)}$ are cell and internal states in the LSTM cell, and $f_\theta$ is a fully-connected layer parameterized by $\theta$. $\odot$ represents element-wise multiplication. In summary, recurrent cellular attention module couples the LSTM and cellular self-attention module to estimate the interaction from neighbor agents using the current hidden states and changes the future dynamics of the center agent to resemble the dynamics of important (i.e., high attention score) neighboring agents.

## 4 EXPERIMENTAL RESULT

**Experimental Setting** The training dataset has the fixed dimension of $64 \times 64$ with RGB channels. The three datasets include 700 chunks (train) and 300 chunks (test). Each chunk has 60 frames for the forest fire and 30 frames for the others. Note, $t_{n_{\text{obs}}}$ is 10 frames in common, and $t_{n_{\text{pred}}}$ is 60 frames (forest fire) or 30 frames (others). The ground truth is a certain state of interest (i.e., binary prediction), such as burning (fire or ember, forest fire), healthy (host-pathogen), and buying (stock market). We use F1 score and area under ROC curve (AUC) as evaluation metrics. Our F1 score is based on the probability threshold 0.5, and AUC assumes that we can have the different threshold for each prediction timestep. Both metrics are only measured in the prediction window ($t_{n_{\text{pred}}} - t_{n_{\text{obs}}}$) and averaged by the number of predicted frames. The training loss in RA-NCA and other baseline NCA networks is based on binary cross-entropy loss. We set 300~1000 epochs, $1e^{-4} \sim 5e^{-4}$ learning rates, batch size 4 chunks and use adam optimizer.

### 4.1 COMPARISON WITH OTHER NCA NETWORKS

As the prevalent form of NCA is based on CNN, we modify the ConvLSTM to preserve the spatial dimension throughout the latent space and name it ConvLSTM-CA (Shi et al., 2015). The ConvLSTM-CA is designed with two convolution layers with a $3 \times 3$ kernel and $1 \times 1$ kernel for encoding and decoding, respectively, and one ConvLSTM cell between them with a $3 \times 3$ kernel. Recently, self-attention based NCA architecture has been proposed (Tesfaldet et al., 2022). To compare it with our model and also to validate the efficacy of recurrent neural networks in RA-NCA, we design another NCA network,

Table 1: Summarized property of compared NCA networks (PI: Permutation Invariance)

| Method | Paramters | Memory | PI |
|---|---|---|---|
| ConvLSTM-CA | 21,313 | ✓ | |
| Attention-CA | 10,617 | | ✓ |
| **Ours** (RA-NCA) | 19,065 | ✓ | ✓ |

named Attention-CA, by removing LSTM from our model, which is fundamentally analogous to the previous work. Experimental results directly from the recent work (Tesfaldet et al., 2022) is available in Appendix C. Also, the results on larger Attention-CA with 20k parameters are provided in Appendix C.

**Data Efficient Deterministic Interaction Learning** We first explore the deterministic forest fire models with the three different forest configurations. Figure 4(a-c) shows the frame-wise mean F1 score depending on the various amount of training data. Note, the training is only performed in the dense forest since the prediction networks should be generalizable across the different configurations if the interaction rules themselves are identical. We observe that if the full amount of training data is given, the performance of the three networks are similar. ConvLSTM-CA shows promising results until 5% of training data particularly in Gaussian Forest, which might be because it involves a non-uniform forest pattern that can be effectively captured by CNN. However, a clear degradation is observed in both Attention-CA and ConvLSTM-CA at the 1% case (training with 7 chunks), while RA-NCA is more robust.

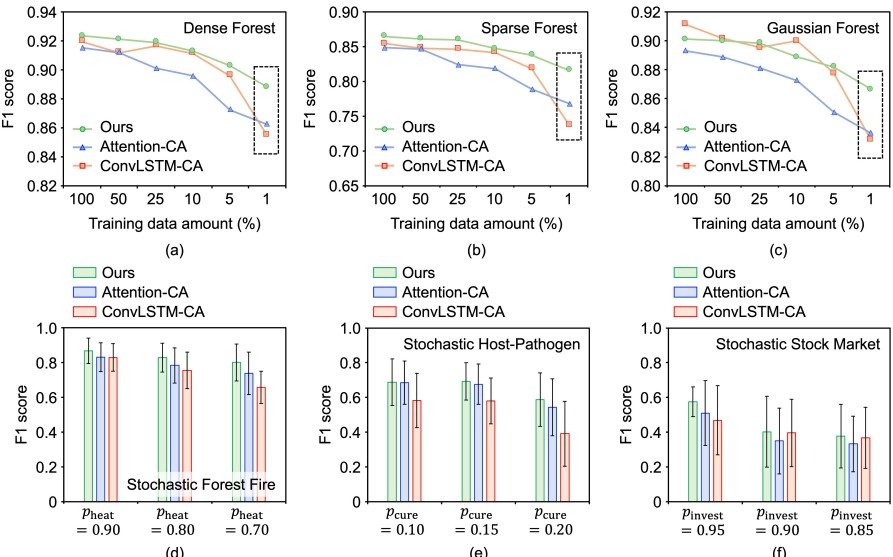

Figure 4: Prediction performance of NCA networks in deterministic and stochastic environments. In the deterministic environment (a-c), the NCA networks are trained on Dense Forest ($p_{\text{heat}} = 1$) with the different amount of training data and then evaluated on the (a) Dense forest, (b) Sparse Forest, and (c) Gaussian Forest. In the stochastic environments (d-f), the NCA networks are trained on the stochastic (d) forest fire, (e) host-pathogen, and (f) stock market models with three different stochasticity levels given 1% amount of training data. The bar and line indicates the mean and standard deviation.

**Data Efficient Stochastic Interaction Learning**   We perform the data efficient learning with 1% of training data in the three datasets with a variable stochasticity. Figure 4(d-f) shows the frame-wise mean F1 score in stochastic forest fire, host-pathogen, and stock market models. Interestingly, the performance disparity between RA-NCA and the others increase in more stochastic settings except for stock market models with high $p_{\text{invest}}$. In particularly, the F1 score of ConvLSTM-CA in forest fire model with $p_{\text{heat}} = 0.70$ and host-pathogen model with $p_{\text{cure}} = 0.20$ is highly degraded, while it performs well with the full training data (see Appendix C). RA-NCA generally exhibits higher mean accuracy than the baselines but sometimes higher variance as well. It indicates, RA-NCA generally provides better predictions than the baselines, but in those cases (particularly AUC), there could be some overlaps with the baselines in the accuracy distribution, as described in the figure. (see the detailed results, including F1 score and AUC, in Appendix C).

**Discussion on Data Efficiency of RA-NCA**   The remaining question is why RA-NCA outperforms the baseline networks in extremely data-limited scenarios. The empirical results have shown that the combination of memory module and self-attention is superior than solely using either of them, such as self-attention without memory in Attention-CA and memory without self-attention in ConvLSTM-CA. Hence, we discuss the performance improvement considering the role of each modules. Figure 4 shows that Attention-CA exhibits the worst F1 score among the three NCA networks except for 1% training data. The poor performance can be resulted from the lack of memory function in the network. However, Attention-CA starts to perform better than ConvLSTM-CA at 1% training data in the deterministic forest fire and most of the stochastic settings as shown in the previous figures. It indicates that the performance gain from self-attention is now higher than the memory function of ConvLSTM-CA. In summary, the better performance of RA-NCA can be resulted from two factors; the memory function relieves the bottleneck of learning the dynamic interaction in Attention-CA, and the permutation invariance allows well-balanced interaction learning even with limited training data than ConvLSTM-CA.

**Scalability: Evaluation in Large Stochastic Systems**   Considering the elementary computation of RA-NCA is defined on 3×3 cells, we can test the trained network regardless of the system scales.

Table 2: Prediction in large systems with RA-NCA networks trained in small systems in stochastic environments. 64→64 and 64→512 indicate the dimension of systems for training and testing. Both the full amount of training data (top four rows) are the 1% amount of training data (bottom four rows) are summarized. The table element indicates mean and standard deviation.

| Method | Stochastic Forest Fire | | Stochastic Host-Pathogen | | Stochastic Stock Market | |
|---|---|---|---|---|---|---|
| (Train→Test) | ($p_{\text{heat}} = 0.90$) | | ($p_{\text{cure}} = 0.15$) | | ($p_{\text{invest}} = 0.95$) | |
| | F1 ↑ | AUC ↑ | F1 | AUC | F1 | AUC |
| RA-NCA (64→64) | 0.8987±0.0598 | 0.9873±0.0152 | 0.7109±0.1065 | 0.9578±0.0308 | 0.6158±0.1745 | 0.7789±0.0842 |
| RA-NCA (64→512) | 0.9172±0.0196 | 0.9878±0.0079 | 0.7241±0.0403 | 0.9593±0.0220 | 0.6177±0.1467 | 0.7758±0.0706 |
| RA-NCA (48→512) | 0.9163±0.0187 | 0.9872±0.0085 | 0.7196±0.0427 | 0.9585±0.0225 | 0.6175±0.1467 | 0.7758±0.0705 |
| RA-NCA (32→512) | 0.9114±0.0215 | 0.9852±0.0088 | 0.7161±0.0422 | 0.9579±0.0227 | 0.6112±0.1501 | 0.7756±0.0705 |
| RA-NCA (64→64) | 0.8672±0.0729 | 0.9768±0.0220 | 0.6957±0.1078 | 0.9443±0.0405 | 0.5762±0.1887 | 0.7751±0.0855 |
| RA-NCA (64→512) | 0.8921±0.0194 | 0.9778±0.0109 | 0.7061±0.0446 | 0.9463±0.0275 | 0.5775±0.1633 | 0.7728±0.0714 |
| RA-NCA (48→512) | 0.8782±0.0228 | 0.9728±0.0127 | 0.6746±0.0531 | 0.9215±0.0269 | 0.5422±0.1742 | 0.7704±0.0729 |
| RA-NCA (32→512) | 0.8851±0.0217 | 0.9635±0.0132 | 0.6262±0.0595 | 0.9239±0.0395 | 0.5660±0.1617 | 0.7693±0.0728 |

The intuition here is that, as long as the interaction rules are same, the prediction quality should be preserved even if the scale of systems significantly alters (*i.e.*, scalability). We perform comprehensive experiments for the scalability in the three training scales of 32×32, 48×48, and 64×64 and the large test scale of 512×512. Table 2 (top four rows) showcases the F1 score and AUC across the three stochastic environments with the full amount of training data. Interestingly, we observe that the accuracy is not significantly different in the three models. The results indicate that RA-NCA successfully learns the interaction rule. Visualization results are provided in Appendix B.

We further investigate the scalability in the data-limited scenarios. We perform the same experiment with 1% training data. The results are provided in Table 2 (bottom four rows). Now, the accuracy disparity across the different training scales is not marginal. The degradation induced by 100%→1% training data is more obvious in the lower scales, particularly in AUC. From this result, we view the data efficiency and system scales are correlated. For example, there are more chances for the model to observe agent interactions in larger scales. Ideally, 1% training data on 64x64 will involve approximately 4 times more agent interactions than that of 32x32.

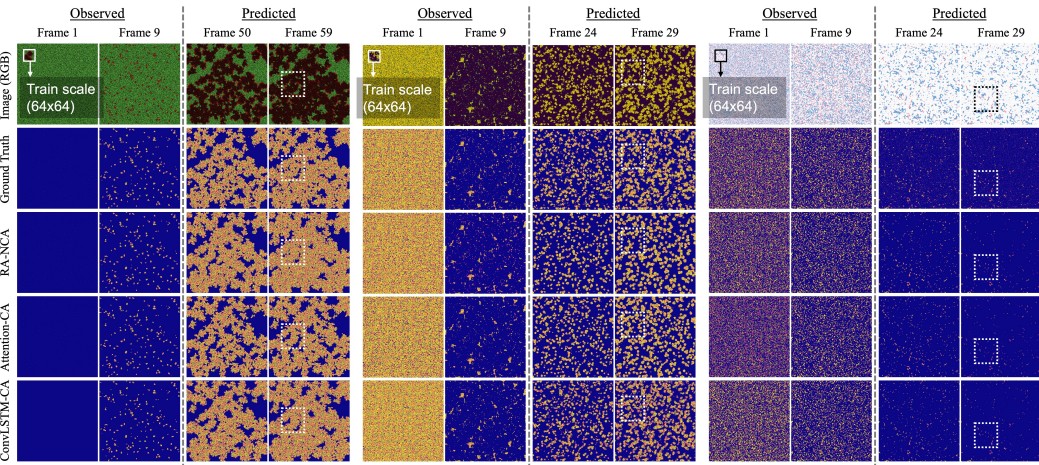

Figure 5: Visualization of prediction in RA-NCA and the baselines on the large stochastic forest fire with $p_{\text{heat}} = 0.90$ (left) host-pathogen model with $p_{\text{cure}} = 0.15$ (middle), and stock market with $p_{\text{invest}} = 0.15$ (right). Training in 64×64 scales with 1% data and evaluation in 512×512 scales. White boxes are marked on the last prediction for the comparison.

Finally, RA-NCA and the baselines are compared in the large systems given the data-limited scenarios. Similarly, the models are trained at 64×64 with the 1% training data and then evaluated at 512×512 in the three stochastic environments. Figure 5 displays the prediction results of the three models with corresponding input and ground truth. The three models are generally scalable, as all of them are NCA-based, and aligned with the ground truth. However, the difference is primarily

Table 3: Prediction in large systems with the baselines trained in small systems in stochastic environments. The amount of training data is 1%.

| Method | Stochastic Forest Fire ($p_{\text{heat}} = 0.90$) | | Stochastic Host-Pathogen ($p_{\text{cure}} = 0.15$) | | Stochastic Stock Market ($p_{\text{invest}} = 0.95$) | |
|---|---|---|---|---|---|---|
| | F1 ↑ | AUC ↑ | F1 | AUC | F1 | AUC |
| RA-NCA | **0.8921±0.0194** | **0.9778±0.0109** | **0.7061±0.0446** | **0.9463±0.0275** | **0.5775±0.1633** | **0.7728±0.0714** |
| Attention-CA | 0.8588±0.0269 | 0.9730±0.0157 | 0.6903±0.0487 | 0.9438±0.0249 | 0.5104±0.1668 | 0.7667±0.0699 |
| ConvLSTM-CA | 0.8638±0.0225 | 0.9708±0.0161 | 0.5935±0.0672 | 0.8982±0.0511 | 0.4663±0.1825 | 0.7545±0.0759 |

observed in the detailed prediction, as marked in the white boxes in the figure. Table 3 showcases that RA-NCA still outperforms the ConvLSTM-CA and Attention-CA in the large scale systems.

## 4.2 COMPARISON WITH VIDEO PREDICTION NETWORKS

Since our problem can be also considered a specific type of image prediction tasks, where modelling the pixel-level dynamics is crucial, a comprehensive comparison between RA-NCA and video prediction networks is performed. Various video prediction networks, including RNN-based (Wang et al., 2017; 2018), GAN-based (Wang et al., 2020), and CNN-based (Gao et al., 2022) architectures. Similarly, we explore the three stochastic environments with $p_{\text{heat}} = 0.90$, $p_{\text{cure}} = 0.15$, and $p_{\text{invest}} = 0.95$. In order to preserve the training procedure of prior works, the baselines are first trained to predict RGB images and then segment the predicted images into binary-state maps by comparing the color norm with the discrete ground truth colors of agent states. The performance disparity between RA-NCA and the baseline networks is noticeable particularly in the data-limited scenarios. Figure 6 shows the frame-wise mean and variance of F1 score in the stochastic environments with 100% and 1% training data. While the video prediction networks are significantly degraded in the 1% training data setting, RA-NCA exhibits negligible degradation in all the three datasets. In summary, RA-NCA provides another perspective to process this type of sequential images (governed by pixel-level dynamics) by treating each of pixels as locally interacting agents. Detailed results are provided in the Appendix D. We also observe that the video prediction networks can generate un-physical predictions (see Appendix D).

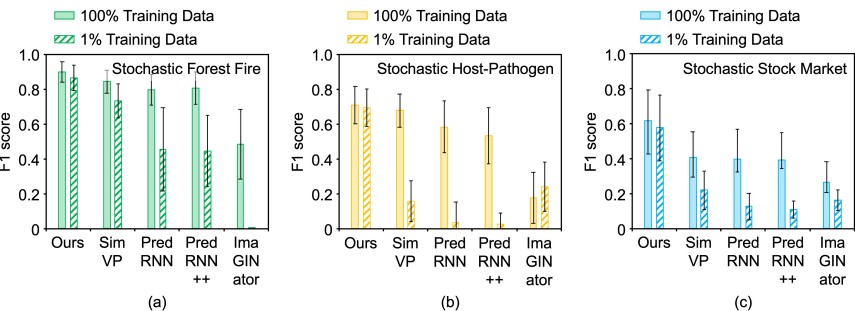

Figure 6: Data efficiency of video prediction networks and RA-NCA in (a) stochastic forest fire, (b) host-pathogen, and (c) stock market models. The bar and line indicate the mean and standard deviation.

## 5 CONCLUSION

We present **R**ecurrent **A**ttention-based **N**eural **C**ellular **A**utomata (RA-NCA) to discover the stochastic local interaction in the multi-agent systems using the temporal and permutation-invariant learning modules. Our experimental results support the key advantages of RA-NCA, the data efficient learning even in the presence of stochastic interactions and generalization on the large-scale systems. We believe the paper provides an useful empirical background in the predictive modeling of locally-interacting multi-agent systems.

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

# A DATASET

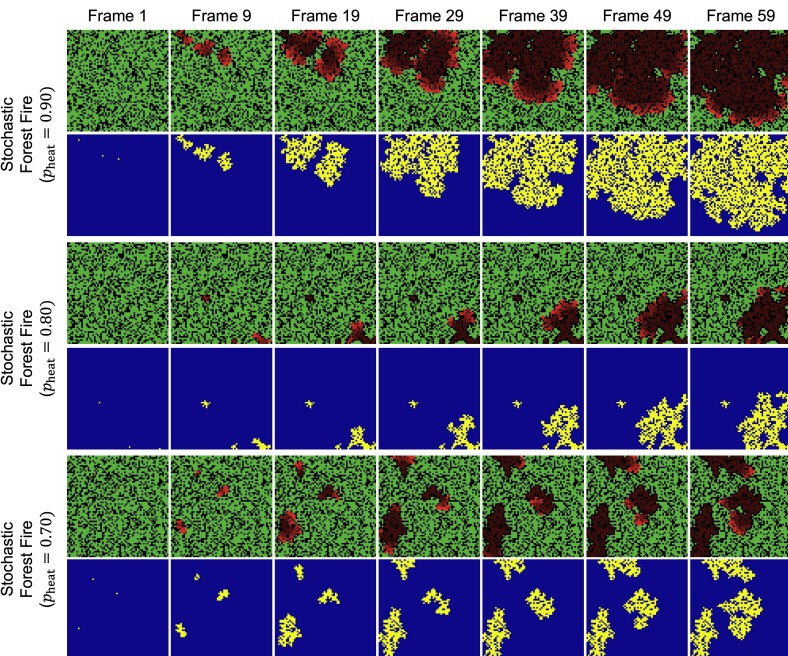

Figure 7: Stochastic forest fire with various $p_{\text{heat}}$ values. First row is RGB-based observation, which is given to the prediction networks as input (black: empty, green: tree, brown: ember, and red: fire). The next row is the corresponding binary ground truth (fire or ember: 1, others:0)

**Forest Fire model**    Forest Fire model is a well-known example of self-organizing complex system. We consider the model as a multi-agent system where four different types of agents (i.e., empty, tree, ember, and fire) are locally interacting. Our forest fire implementation is based on NetLogo, which is a widely used agent-based modeling platform (Tisue & Wilensky, 2004). We modify the interaction rule to be non-trivial based on the prior works (Rothermel, 1972; Chen et al., 1990).

There are few parameters that primarily control the evolution of fire in the model; $q_{\text{seed}}$, $q_{\text{threshold}}$, $q_{\text{die}}$, and $q_{\text{transfer}}$. The parameters are closely related to the agent's internal state referred to as a "heat" value ($q$). $q_{\text{seed}}$ is the heat value of initial fire seeds. $q_{\text{threshold}}$ is a threshold value that defines the transition from tree agents to fire agents. $q_{\text{transfer}}$ controls the efficiency of heat accumulation for tree agents from neighboring fire and ember agents (i.e., $q_{(i,j)}(t+1) = q_{(i,j)}(t) + q_{\text{transfer}} \sum_{S_{(i,j)}} q_s(t)$) where $S_{(i,j)}$ is a set of neighboring agents from location $(i,j)$ and $s$ is an element of the set. After the heat interaction between agents are executed, $q_{\text{die}}$ defines the speed of cooling down of fire and ember agents (i.e., $q(t+1) = q(t) - q_{\text{die}}$). Once fire agents experience the decrease in the heat value, it darkens its color. It takes 6 frames to become ember from fire and 12 frames to no longer transfer heat to neighboring agents.

The forest fire model evolves following the sequence: **1)** Create 64×64 agent map filled with randomly distributed tree and empty agents. **2)** Define 3 fire seeds with $q_{\text{seed}}$ at random locations. **3)** Each tree agent accumulates the heat from neighboring fire and ember agents using the equation related to $q_{\text{transfer}}$. **4)** Check if the current heat value exceeds $q_{\text{threshold}}$. **5)** Decrease the heat values of fire and ember agents using the equation related to $q_{\text{die}}$. Repeat from **3)** to **5)**.

For the parameter values, we set $q_{\text{seed}} = 6$, $q_{\text{threshold}} = 3$, $q_{\text{die}} = 1$, and $q_{\text{transfer}} = 0.3$. In stochastic forest fire, we introduce another variable $p_{\text{heat}}$, indicating the probability of accumulating the heat from each of neighboring agents. It is applied to the equation related to $q_{\text{transfer}}$ within a Bernoulli

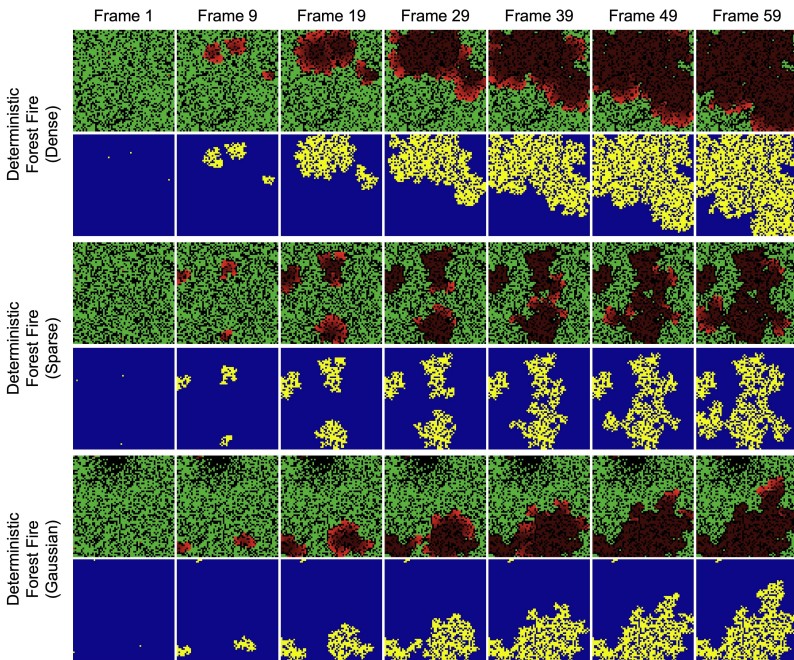

Figure 8: Deterministic forest fire with different forest configurations.

random variable (i.e., $q_{(i,j)}(t+1) = q_{(i,j)}(t) + q_{\text{transfer}} \sum_{S_{(i,j)}} \text{Bernoulli}(p_{heat})q_s(t)$). Our goal is to predict fire and ember agents in the models with various $p_{\text{heat}}$ values. Figure 7 shows the example scenes of the system with the three different $p_{\text{heat}}$ values. In deterministic forest fire, We design the three different forest configuration; dense forest, sparse forest, and gaussian forest. The dense forest is initially filled with 65% tree agents and 35% empty agents. The sparse forest is initially filled with 60% tree agents and 40% empty agents. The spatial distribution of tree agents in dense and sparse forest follow random uniform. We observe that lower than 60% makes the fire propagation vanishes too early and higher than 65% makes the fire front too trivial (e.g., radial diffusion). For gaussian forest, we first create dense forest and then remove tree agents following two-dimensional gaussian distribution with random mean from (-1,-1) to (1,1) and random variance $\in [0.05, 0.1]$ assuming the forest coordinate is normalized to (-1,1). Figure 8 shows the three forest configurations. We generate 300 test chunks for the dense forest and 100 test chunks for the others.

**Host-Pathogen model**  Host-pathogen model is originated from the theoretical understanding of the interaction between viruses and host organisms in a population level. It can be also considered a two-dimensional version of prey-predator model, where two types of agents in an adversarial relationship compete each other. We use the prior implementation developed by PyCX (Sayama, 2013), which is a python-based complex system simulator. Our host-pathogen model has four different types of agents (i.e., empty, dead, healthy, and infected) with local and stochastic interactions.

The host-pathogen model evolves following the sequence: **1)** Create 64×64 agent map filled with randomly distributed infected (1%), healthy (75%), and empty (the rest) agents. **2)** For dead agents, cured by each of neighboring healthy agents with the probability $p_{\text{cure}}$. **4)** For healthy agents, infected by each of neighboring infected agents with probability $p_{\text{infect}}$. **5)** For infected agents, transitioned to dead agents. Repeat from **2)** to **5)**.

We observe that if the agent map is full of healthy agents, the evolution of systems become trivial (radial diffusion) depending on the probability. To avoid such scenarios, we introduce additional empty agents which are not interacting with any other agents (i.e., inactive cells). We fix $p_{\text{infect}} = 0.85$, which is the default value in the implementation, and change $p_{\text{cure}}$. Our goal is to predict healthy agents in the models with various $p_{\text{cure}}$ values. Figure 9 shows the example scenes of the system with the three different $p_{\text{cure}}$ values.

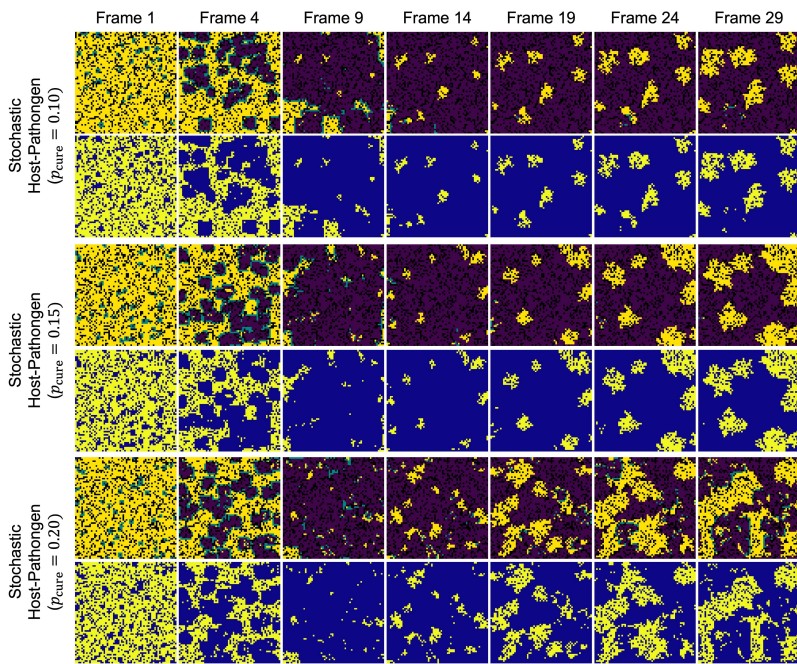

Figure 9: Stochastic host-pathogen model with various $p_{\text{cure}}$ values. First row is RGB-based observation (green: infected, yellow: healthy, purple: dead, and black: empty). The next row is the corresponding binary ground truth (healthy: 1, others:0)

Table 4: Transition matrix in the stock market model. "Neighbor" in the first row indicates the most used action by neighboring agents.

| Neighbor | Next Action | | |
| --- | --- | --- | --- |
| | Buying | Holding | Selling |
| Buying | $p_{\text{invest}} + M$ | $0.5(1 - p_{\text{invest}} - M)$ | $0.5(1 - p_{\text{invest}} - M)$ |
| Holding | $(1 - p_{\text{invest}})(0.5 + 0.5M)$ | $p_{\text{invest}}$ | $(1 - p_{\text{invest}})(0.5 - 0.5M)$ |
| Selling | $(1 - p_{\text{invest}})(0.5 + 0.5M)$ | $(1 - p_{\text{invest}})(0.5 - 0.5M)$ | $p_{\text{invest}}$ |

**Stock Market model**   The complexity of investor's behavior in the stock market has been studied through cellular automata in the prior work (Wei et al., 2003). They aimed to model the "imitation" behavior of investors affected by neighboring investors, which often makes the stock market unstable and complex. Reversely, we aim to predict this system assuming the unknown stochastic interaction between investors (i.e., agents). There are four different types of actions in the agent (i.e., hold, sell, buy, and inactive). The inactive action is defined by ourselves to prohibit too repeated buying actions. Table 4 shows the probabilistic transition matrix in the positive market scenario, which is already given in the paper (Wei et al., 2003).

The stock market model evolves following the sequence: **1)** Create 64×64 agent map filled with randomly chosen actions (e.g., buy, sell, hold). **2)** Each of agents makes the stochastic action based on neighboring agents. **3)** The agents made two consecutive buying actions become inactive. **4)** The inactive agents in the previous sequence do the buying action. Repeat from **2)** to **4)**.

There is two variables $p_{\text{invest}}$ and $M$ to control the system. We set $M = 0.05$ (i.e., the market is positive) and try various $p_{\text{invest}}$ values. We introduce the inactive state to make the system more dynamic. Note, the basic implementation without the inactive state can quickly make the system static (no more evolution) with clustered agents. Our goal is to predict agents with the buying action in the models with various $p_{\text{invest}}$ values. Figure 10 shows the example scenes of the system with the three different $p_{\text{invest}}$ values.

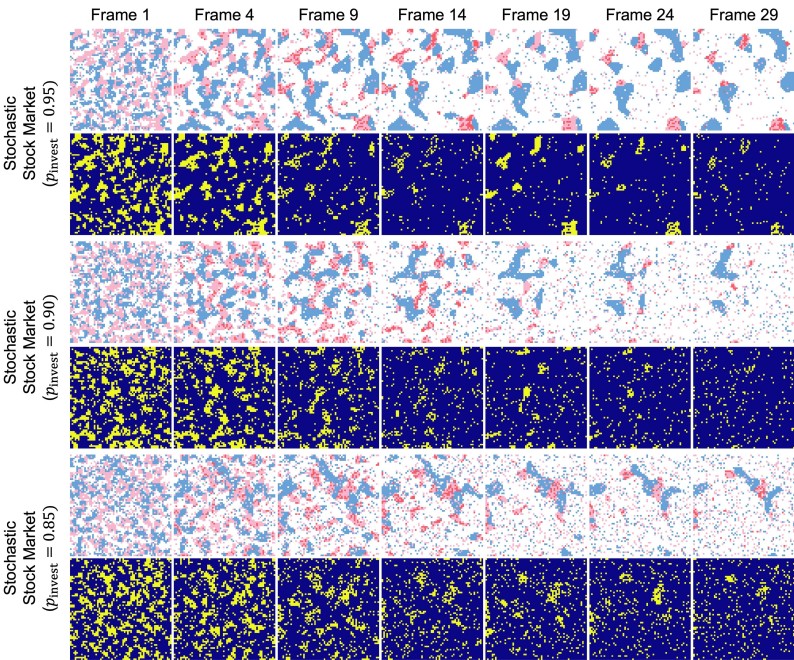

Figure 10: Stochastic stock market model with various $p_{\text{invest}}$ values. First row is RGB-based observation (white: hold, blue: sell, pink: buy, and red: inactive). The next row is the corresponding binary ground truth (buy: 1, others:0)

Table 5: Comparison in RA-NCA networks with various encoding dimensions. Compared encoding dimensions ($n$) are 16 (6,185 parameters), 32 (19,065 parameters), and 48 (39,113 parameters). The networks are trained with 1% training data.

| Method | Forest Fire | | Host-Pathogen | | Stock Market | |
|---|---|---|---|---|---|---|
| (Parameters) | ($p_{\text{heat}} = 0.90$) | | ($p_{\text{cure}} = 0.15$) | | ($p_{\text{invest}} = 0.95$) | |
| | F1 ↑ | AUC ↑ | F1 | AUC | F1 | AUC |
| RA-NCA ($n = 16$) | 0.8335±0.0934 | 0.9696±0.0292 | 0.6810±0.1160 | 0.9287±0.0479 | 0.5560±0.1909 | 0.7738±0.0852 |
| RA-NCA ($n = 32$) | **0.8672±0.0729** | **0.9768±0.0220** | 0.6957±0.1078 | 0.9443±0.0405 | 0.5762±0.1887 | 0.7751±0.0855 |
| RA-NCA ($n = 48$) | 0.8636±0.0729 | 0.9758±0.0224 | **0.7001±0.1087** | **0.9485±0.0372** | **0.5848±0.1821** | **0.7756±0.0852** |

# B  ANALYSIS ON RA-NCA NETWORK

**Various Encoding Dimensions**  We explore other encoding dimensions of RA-NCA. The encoding dimension ($n$) is associated across the feature extractor ($R^3 \to R^n$), Recurrent cellular attention module ($R^n \to R^n$), and the decoder ($R^n \to R^1$) in RA-NCA. We consider three encoding dimensions, $n = 16$, $n = 32$, and $n = 48$, in stochastic forest fire, host-pathogen, and stock market models with 1% amount of training data. The results are given in Table 5. We observe relatively higher accuracy gap between $n = 16$ and $n = 32$ models, than between $n = 32$ and $n = 48$ models. It implies that the limited capacity of n=16 model is relieved from $n >= 32$ models. From this result, we choose $n = 32$ as a baseline dimension (compact but not too small capacity). The $n = 48$ model is worse than the n=32 model in the stochastic forest fire, but better in the host-pathogen and stock market datasets. It might be because 1% training data is too less compared to the larger capacity of the $n = 48$ model, particularly in the forest fire dataset. Note, the required amount of training data to avoid the degradation will be different in the three environments since the models predict 50 frames in the forest fire, while predict 20 frames in the host-pathogen and stock market.

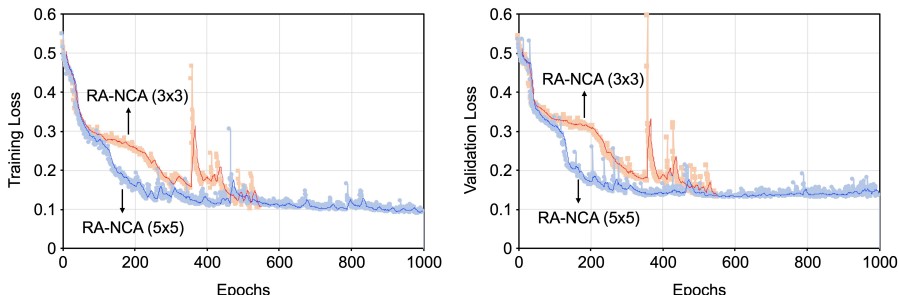

Figure 11: Training dynamics of RA-NCA with 3 ×3 operation (Moore's neighborhood) and 5×5 operation. Both the models are trained for 1000 epochs. The loss curve of RA-NCA (3×3) is omitted from 550 epochs since it starts to diverge. The training BCE loss (left) and validation BCE loss (right) are reduced to the similar level in both the models.

**Different Neighborhood Size in RA-NCA**  We investigate another scale of 5x5 neighborhood in RA-NCA. The training and evaluation environment is the deterministic forest fire model. The interaction rules in the dataset remain the same. The full amount of training data is employed, and both the training and test environments are Dense Forest. It is important to mention that the 5×5 model does not know about the interaction scale in the forest fire dataset. Figure 11 shows the training dynamics (BCE loss) of the two models. We observe that the training speed is different in the two models, but their minimum validation loss is converged into the similar level. Table 6 compares the F1 score and AUC of the two models, and the neighborhood scale of 5×5 does

Table 6: Comparison of RA-NCA networks with different neighborhood size. RA-NCA (3×3) is the model in the main text, RA-NCA (5×5) operates on 5×5 cells.

| Method | Deterministic Forest Fire | |
|---|---|---|
| | F1 ↑ | AUC ↑ |
| RA-NCA (3×3) | 0.8880±0.0610 | 0.9766±0.0219 |
| RA-NCA (5×5) | 0.8786±0.0592 | 0.9787±0.0195 |

Table 7: Comparison of RA-NCA networks with different recurrent neural networks. RA-NCA (LSTM) refers the proposed architecture in the main text, and RA-NCA (RNN) replaces the LSTM module to the basic RNN module. RA-NCA (LSTM) and RA-NCA (RNN) involve 19,065 and 12,729 parameters, respectively. The baseline model (Attention-CA) is also included. The networks are trained with 1% training data.

| Method | Stochastic Forest Fire ($p_{\text{heat}} = 0.90$) | | Stochastic Host-Pathogen ($p_{\text{cure}} = 0.15$) | | Stochastic Stock Market ($p_{\text{invest}} = 0.95$) | |
|---|---|---|---|---|---|---|
| | F1 ↑ | AUC ↑ | F1 | AUC | F1 | AUC |
| Attention-CA | 0.8305±0.0822 | 0.9705±0.0283 | 0.6759±0.1159 | 0.9433±0.0372 | 0.5103±0.1862 | 0.7692±0.0829 |
| RA-NCA (RNN) | 0.8410±0.0791 | 0.9725±0.0266 | 0.6547±0.1199 | 0.9373±0.0376 | 0.5692±0.1857 | **0.7756±0.0848** |
| RA-NCA (LSTM) | **0.8672±0.0729** | **0.9768±0.0220** | **0.6957±0.1078** | **0.9443±0.0405** | **0.5762±0.1887** | 0.7751±0.0855 |

not degrade the performance (lower F1 but higher AUC (ROC)). However, we would like to also mention that 3x3 Moore's neighborhood is the most common configuration in Cellular Automata, and hence neural CA architectures are generally aimed to perform on the 3x3 Moore's neighborhood (Mordvintsev et al., 2020; Hernandez et al., 2021; Gilpin, 2019).

**RNN-based RA-NCA network** We investigate the Attention+RNN model. Its design is basically similar with RA-NCA, but we replace the LSTM module (Figure 3(b) right) with the basic RNN module. The hidden state in the RNN module is defined as follows:

$$\mathbf{h}_t = \tanh(\mathbf{W}_{ih}\mathbf{x}_t + \mathbf{b}_{ih} + \mathbf{W}_{hh}\mathbf{h}_{t-1} + \mathbf{b}_{hh}) \tag{7}$$

The training environments are stochastic and data-limited forest fire, host-pathogen, and stock market models with 1% training data. The comparison is provided in Table 7. Their parameters are not exactly matching each other (10k for Attention-CA, 12k and 20k for RA-NCA (RNN) and RA-NCA (LSTM)) since we aim to understand the efficacy of the memory module in the same architecture. The F1 score and AUC are generally higher in both the RA-NCA models except for the host-pathogen. Our primary observation is that the accuracy gap between Attention-CA and RA-NCA (LSTM) is higher in the forest fire and stock market, indicating the memory function (LSTM) plays an important role in these environments than the host-pathogen. This might be because the forest fire model and stock market involve more complex interactions, such as dynamic heat dissipation and accumulation (forest fire) and dynamic inactive state (stock market). In this light, introducing the memory function will easily improve the accuracy in the forest fire and stock market. However, as RNN is often unstable due to the lack of cell states, this may make the prediction rather unstable in less dynamic systems. In summary, introducing the memory function is generally helpful if the systems involve the complex behaviors. Also, considering AUC of RA-NCA (RNN) in marginally higher than that of RA-NCA (LSTM), we can say that RA-NCA (LSTM) generally performs better than the two models.

**Visualization of RA-NCA Prediction** We attach the visualization of prediction results on the large systems (512×512) from RA-NCA trained on the small systems (64×64) with the full amount of training data. Figure 12, 13, 14 are the visualization of the large forest fire ($p_{\text{heat}} = 0.90$), large host-pathogen ($p_{\text{cure}} = 0.15$), and large stock market ($p_{\text{invest}} = 0.95$) models, respectively.

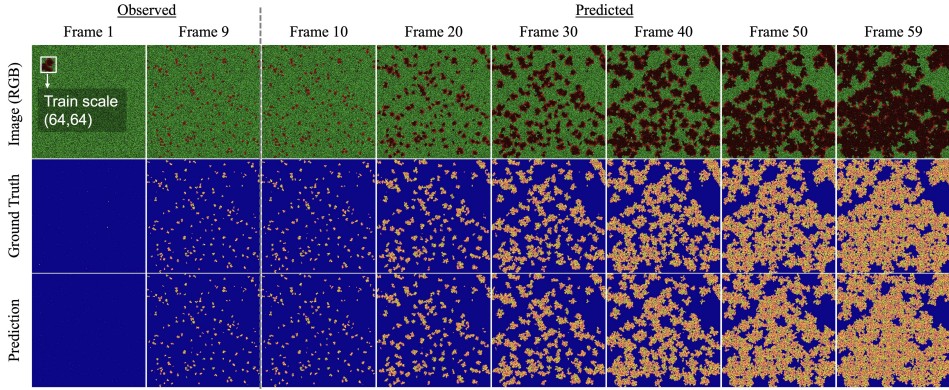

Figure 12: Prediction results of RA-NCA in the large-scale stochastic forest fire model ($p_{\text{heat}}$ = 0.90). Training on the 64×64 agent maps and testing on the 512×512 agent maps.

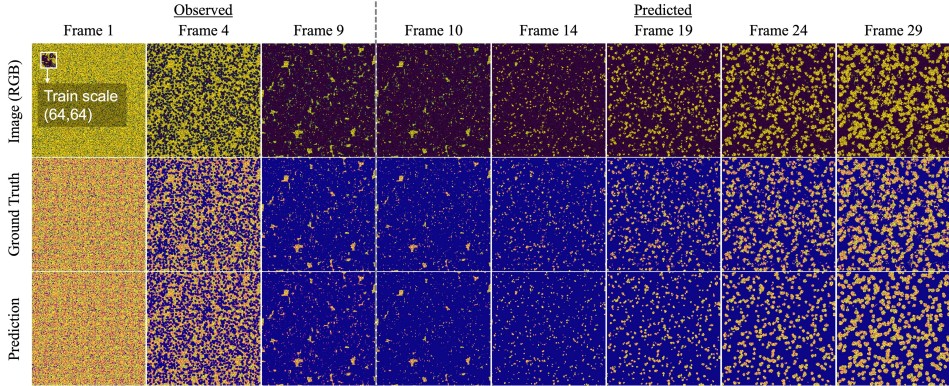

Figure 13: Prediction results of RA-NCA in the large-scale stochastic host-pathogen model ($p_{\text{cure}}$ = 0.15). Training on the 64×64 agent maps and testing on the 512×512 agent maps.

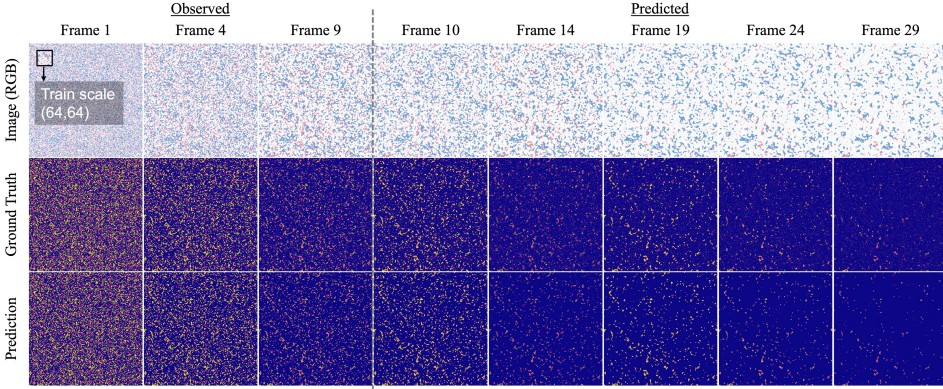

Figure 14: Prediction results of RA-NCA in the large-scale stock market model ($p_{\text{invest}}$ = 0.95). Training on the 64×64 agent maps and testing on the 512×512 agent maps.

## C COMPARISON WITH OTHER NCA NETWORKS

**Challenges in Employing prior NCA networks**    Most NCA networks, including the state-of-the-art Attention-Transformer NCA (Tesfaldet et al., 2022), aim to update cells towards a stable fixed-point. For example, the NCA model receives an input image, returns an updated image, and then utilizes it as the next input image. It repeats this process by 8∼32 steps and generate almost no updates after 32 steps. In image denoising, once a noisy image is fully denoised, the model should not update pixels anymore (otherwise causing a divergence). Hence, ensuring a fixed-point is an important goal as it addresses stationary data. However, it poses a challenge in predicting dynamical systems, where the model should generate a sequence. In other words, after the model generate an image at time $t$, it should be able to continually generate the next images at $t + 1$, $t + 2$, ... , instead of staying at time $t$ treating it as a fixed-point.

Further, these approaches will have high computational complexities in both training and inference. For example, Attention-Transformer NCA requires 8∼32 steps to generate a single image for training and 64 steps for inference. For the forest fire model, where the model needs to generate 50 prediction images, the computational graph to calculate its gradient can be very long (*i.e.*, up to 32×50 (=1600) sequences). It significantly increases the GPU memory and training time as well. Similarly, the model should repeat the update steps up to 64×50 (=3200) times for inference, which still requires a very high GPU memory and takes a long inference time.

To illustrate the preceding challenges, we adopted Attention-Transformer NCA to the deterministic forest fire model with full training data. Due to the abovementioned computational difficulties, we set the prediction timesteps to 10 (observe 10 frames and predict 10 frames). For the implementation, we employ the codebase attached in the supplement of the paper (https://openreview.net/forum?id=9t24EBSlZOa). We modify the model size (23,731 parameters) to be similar with our RA-NCA by setting the embedding dimension to 64 and MLP dimension to 16. Figure 15 and 16 demonstrate the training dynamics of the model and showcases the failure case in predicting the deterministic forest fire.

In summary, our RA-NCA follows the common architectural strategy in NCA, operating on the Moore's neighborhood, but our problem cannot be effectively and efficiently solved by the Attention-Transformer NCA model. Also, we would like to mention that the above-mentioned two challenges are applied to the other existing NCA models (non-recurrent) as well, but such discussion has been missed since their applications are focused on the stationary data.

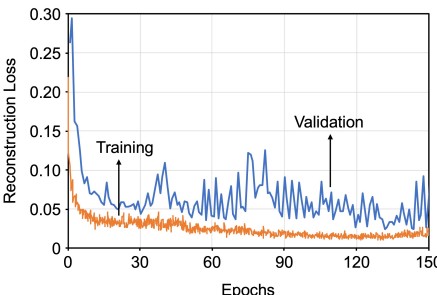

Figure 15: Training dynamics of Attention-Transformer NCA (Tesfaldet et al., 2022) in the deterministic forest fire dataset.

**Large Attention-CA network**    We design a large Attention-CA baseline with 20,297 trainable parameters. Now, the number of parameters in the models is ConvLSTM-CA (21,169), RA-NCA (19,065), Attention-CA (small: 10,617, large: 20,297). We studied the performance of the large

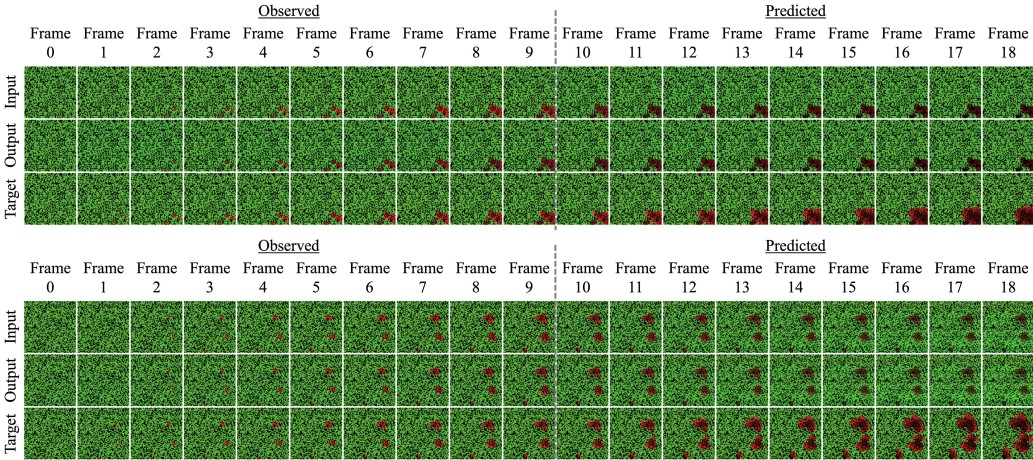

Figure 16: Prediction results of Attention-Transformer NCA (Tesfaldet et al., 2022) on training data (top) and validation data (bottom).

Table 8: Comparison between large and small Attention-CA models. The networks are trained on 1% training data in stochastic forest fire, host-pathogen, and stock market models.

| Method | Stochastic Forest Fire | | Stochastic Host-Pathogen | | Stochastic Stock Market | |
|---|---|---|---|---|---|---|
| (Parameters) | ($p_{\text{heat}} = 0.90$) | | ($p_{\text{cure}} = 0.15$) | | ($p_{\text{invest}} = 0.95$) | |
| | F1 ↑ | AUC ↑ | F1 | AUC | F1 | AUC |
| Attention-CA (10,617) | 0.8305±0.0822 | 0.9705±0.0283 | 0.6759±0.1159 | 0.9433±0.0372 | 0.5103±0.1862 | 0.7692±0.0829 |
| Attention-CA (20,297) | 0.2880±0.1814 | 0.8118±0.1338 | 0.6770±0.1109 | 0.9403±0.0378 | 0.5050±0.1917 | 0.7684±0.0838 |

Attention-CA in stochastic and data-limited settings. The training environments are stochastic forest fire ($p_{\text{heat}} = 0.90$), host-pathogen ($p_{\text{cure}} = 0.15$), stock market ($p_{\text{invest}} = 0.95$) with the 1% amount of training data. As shown in Table 8, the large Attention-CA shows similar or poorer performance compared to the small Attention-CA. In fact, we observe that, the training of large Attention-CA is relatively unstable than the Small Attention-CA, and again showing lower F1 scores and AUC than RA-NCA in the three datasets. This indicates that simply increasing the model complexity of Attention-CA does not help learning the interaction rules in stochastic and data-limited scenarios.

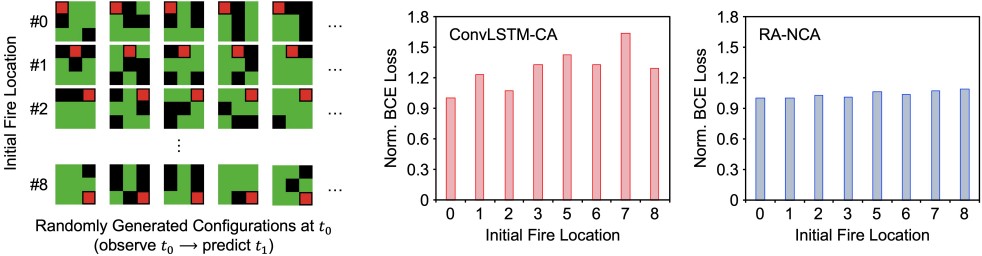

Figure 17: Spatial dependency in RA-NCA and ConvLSTM-CA. The right graphs show the average loss of predicted burning probabilities, which is normalized to make the minimium loss value as 1.

**Spatial Dependency in Interaction Learning** We design an experiment to measure the spatial dependency on the order of neighboring agents in ConvLSTM-CA and RA-NCA. The networks are taken from the pre-trained ones in Figure 4(d) (stochastic forest fire with $p_{\text{heat}} = 0.7$). We consider very small 3×3 forest configurations under the same stochasticity and ignite one neighboring agent. The other cells that are not ignited are randomly chosen to be empty or tree agents. In this setting, we can estimate the prediction performance as a function of the fire seed location. Figure 17 displays the examples of the small forest configurations. We generate ∼1000 simulations for each of the fire seed locations. The networks predict a single frame after observing the first frame so that the

Table 9: Comparison of F1 score with other NCA architectures trained on 100% training data in stochastic forest fire model (top), host-pathogen model (middle), and stock market model (bottom)

| Method | $p_{\text{heat}} = 0.90$ F1 ↑ | $p_{\text{heat}} = 0.80$ F1 | $p_{\text{heat}} = 0.70$ F1 |
|---|---|---|---|
| ConvLSTM-CA | 0.8879±0.0688 | 0.8761±0.0728 | 0.8424±0.0994 |
| Attention-CA | 0.8891±0.0635 | 0.8719±0.0747 | 0.8374±0.0993 |
| **RA-NCA** | **0.8987±0.0598** | **0.8801±0.0737** | **0.8588±0.0928** |

| Method | $p_{\text{cure}} = 0.10$ F1 ↑ | $p_{\text{cure}} = 0.15$ F1 | $p_{\text{cure}} = 0.20$ F1 |
|---|---|---|---|
| ConvLSTM-CA | 0.7040±0.1277 | 0.7022±0.1116 | 0.6072±0.1556 |
| Attention-CA | **0.7110±0.1211** | 0.7048±0.1087 | 0.6008±0.1588 |
| **RA-NCA** | 0.7101±0.1232 | **0.7109±0.1065** | **0.6088±0.1530** |

| Method | $p_{\text{invest}} = 0.95$ F1 ↑ | $p_{\text{invest}} = 0.90$ F1 | $p_{\text{invest}} = 0.85$ F1 |
|---|---|---|---|
| ConvLSTM-CA | **0.6167±0.1747** | **0.4229±0.2031** | **0.3913±0.1812** |
| Attention-CA | 0.5407±0.1865 | 0.3507±0.2041 | 0.3378±0.1615 |
| **RA-NCA** | 0.6158±0.1745 | 0.4195±0.2034 | 0.3895±0.1788 |

Table 10: Comparison of AUC with other NCA architectures trained on 100% training data in stochastic forest fire model (top), host-pathogen model (middle), and stock market model (bottom)

| Method | $p_{\text{heat}} = 0.90$ AUC ↑ | $p_{\text{heat}} = 0.80$ AUC | $p_{\text{heat}} = 0.70$ AUC |
|---|---|---|---|
| ConvLSTM-CA | 0.9864±0.0160 | 0.9858±0.0182 | 0.9823±0.0252 |
| Attention-CA | 0.9860±0.0158 | 0.9865±0.0166 | 0.9867±0.0168 |
| **RA-NCA** | **0.9873±0.0152** | **0.9887±0.0150** | **0.9894±0.0151** |

| Method | $p_{\text{cure}} = 0.10$ AUC ↑ | $p_{\text{cure}} = 0.15$ AUC | $p_{\text{cure}} = 0.20$ AUC |
|---|---|---|---|
| ConvLSTM-CA | 0.9775±0.0179 | 0.9554±0.0322 | 0.9052±0.0632 |
| Attention-CA | 0.9789±0.0173 | 0.9568±0.0312 | 0.9016±0.0648 |
| **RA-NCA** | **0.9795±0.0167** | **0.9578±0.0308** | **0.9062±0.0643** |

| Method | $p_{\text{invest}} = 0.95$ AUC ↑ | $p_{\text{invest}} = 0.90$ AUC | $p_{\text{invest}} = 0.85$ AUC |
|---|---|---|---|
| ConvLSTM-CA | 0.7785±0.0484 | **0.7000±0.0821** | **0.6959±0.0652** |
| Attention-CA | 0.7719±0.0833 | 0.6956±0.0796 | 0.6908±0.0616 |
| **RA-NCA** | **0.7789±0.0842** | 0.6997±0.0820 | 0.6957±0.0654 |

experiment can directly focus on the interaction between a single fire agent to neighbor agents. The graphs in the figure describe that the performance of ConvLSTM-CA is biased to the few certain locations in neighborhood while RA-NCA is well-balanced regardless of the agent locations.

**Data Efficient Interaction Learning**  We summarize the frame-wise mean and standard deviation of F1 score and AUC depending on the amount of training data. Table 9 and 11 show the F1 score comparison with other NCA networks in the stochastic environments trained with 100% and 1% training data, respectively. Table 10 and 12 shows the AUC comparison with other NCA networks in the stochastic environments trained with 100% and 1% training data, respectively. The elements in the tables indicate the mean and standard deviation (*i.e.*, mean±std).

Table 11: Comparison of F1 score with other NCA architectures trained on 1% training data in stochastic forest fire model (top), host-pathogen model (middle), and stock market model (bottom)

| Method | $p_{heat} = 0.90$ F1 ↑ | $p_{heat} = 0.80$ F1 | $p_{heat} = 0.70$ F1 |
|---|---|---|---|
| ConvLSTM-CA | 0.8293±0.0802 | 0.7546±0.1052 | 0.6569±0.0935 |
| Attention-CA | 0.8305±0.0822 | 0.7837±0.1014 | 0.7379±0.1210 |
| **RA-NCA** | **0.8672±0.0729** | **0.8280±0.0825** | **0.8003±0.1067** |

| Method | $p_{cure} = 0.10$ F1 ↑ | $p_{cure} = 0.15$ F1 | $p_{cure} = 0.20$ F1 |
|---|---|---|---|
| ConvLSTM-CA | 0.5834±0.1565 | 0.5793±0.1317 | 0.3916±0.1870 |
| Attention-CA | 0.6855±0.1245 | 0.6759±0.1159 | 0.5440±0.1630 |
| **RA-NCA** | **0.6883±0.1339** | **0.6957±0.1078** | **0.5870±0.1539** |

| Method | $p_{invest} = 0.95$ F1 ↑ | $p_{invest} = 0.90$ F1 | $p_{invest} = 0.85$ F1 |
|---|---|---|---|
| ConvLSTM-CA | 0.4687±0.1991 | 0.3963±0.1946 | 0.3681±0.1764 |
| Attention-CA | 0.5103±0.1862 | 0.3502±0.1895 | 0.3329±0.1595 |
| **RA-NCA** | **0.5762±0.1887** | **0.4025±0.2036** | **0.3782±0.1826** |

Table 12: Comparison of AUC with other NCA architectures trained on 1% training data in stochastic forest fire model (top), host-pathogen model (middle), and stock market model (bottom)

| Method | $p_{heat} = 0.90$ AUC ↑ | $p_{heat} = 0.80$ AUC | $p_{heat} = 0.70$ AUC |
|---|---|---|---|
| ConvLSTM-CA | 0.9666±0.0290 | 0.9462±0.0453 | 0.9473±0.0429 |
| Attention-CA | 0.9705±0.0283 | 0.9679±0.0323 | 0.9703±0.0298 |
| **RA-NCA** | **0.9768±0.0220** | **0.9732±0.0252** | **0.9763±0.0263** |

| Method | $p_{cure} = 0.10$ AUC ↑ | $p_{cure} = 0.15$ AUC | $p_{cure} = 0.20$ AUC |
|---|---|---|---|
| ConvLSTM-CA | 0.9201±0.0681 | 0.8934±0.0649 | 0.8252±0.0958 |
| Attention-CA | 0.9656±0.0235 | 0.9433±0.0372 | 0.8767±0.0755 |
| **RA-NCA** | **0.9698±0.0254** | **0.9443±0.0405** | **0.8878±0.0743** |

| Method | $p_{invest} = 0.95$ AUC ↑ | $p_{invest} = 0.90$ AUC | $p_{invest} = 0.85$ AUC |
|---|---|---|---|
| ConvLSTM-CA | 0.7582±0.0874 | 0.6902±0.0825 | 0.6900±0.0647 |
| Attention-CA | 0.7692±0.0829 | 0.6946±0.0794 | 0.6892±0.0615 |
| **RA-NCA** | **0.7751±0.0855** | **0.6971±0.0821** | **0.6932±0.0655** |

# D    COMPARISON WITH VIDEO PREDICTION NETWORKS

Figure 18: Unnatural prediction results from video prediction model and corresponding results from RA-NCA. Both models observe the first 10 frames (0∼9) and predict the next 50 frames (10∼59).

**Unphysical Prediction in Video Prediction Networks**    We observe that video prediction networks often predict unnatural behavior of agents, which should not be happened if they actually learn the interaction. For example, Figure 18 shows the predicted burning states (e.g., fire for yellow and no fire for blue) of agents in forest fire model and corresponding ground truth. However, PredRNN in the figure Wang et al. (2017) predicts the sparse areas will be burning in later frames, which is not physically possible since there is no tree existing in the forest. Moreover, complex fire front is not appropriately predicted.

A possible reason contributing to the failure will be related to the spatial structures in the latent space, which makes the NCAs clearly different to the video prediction networks. NCA architectures are generally focused on performing on the individual cell, by calculating the update for each of cells, using the 3x3 neighboring information. For example, if input dimension is $\mathbb{R}^{H \times W \times C}$, ($H$, $W$: height and width, $C$: channel), then latent space is $\mathbb{R}^{H \times W \times C'}$, and final output is $\mathbb{R}^{H \times W \times C}$ or $\mathbb{R}^{H \times W \times 1}$ (in our case). In other words, the models perform agent-wise processing, which only changes the channel dimension ($C \rightarrow C' \rightarrow C$ or 1) preserving the spatial dimension ($H \times W - > H \times W \rightarrow H \times W$).

Table 13: Comparison with video prediction networks in deterministic forest fire with various configurations. 100% training data is used.

| Method | Dense Forest (F1↑) | Sparse Forest (F1) | Gaussian Forest (F1) |
|---|---|---|---|
| PredRNN (Wang et al., 2017) | $0.8473\pm0.0710$ | $0.7369\pm0.1102$ | $0.7968\pm0.1448$ |
| PredRNN++ (Wang et al., 2018) | $0.8584\pm0.0692$ | $0.7373\pm0.1357$ | $0.8228\pm0.1071$ |
| ImaGINator (Wang et al., 2020) | $0.5436\pm0.2507$ | $0.3608\pm0.2289$ | $0.4937\pm0.2622$ |
| SimVP (Gao et al., 2022) | $0.8823\pm0.0528$ | $0.7807\pm0.1173$ | $0.8777\pm0.0673$ |
| **RA-NCA** (Ours) | $\mathbf{0.9232\pm0.0468}$ | $\mathbf{0.8648\pm0.0882}$ | $\mathbf{0.9011\pm0.0998}$ |

Table 14: Comparison with video prediction networks in deterministic forest fire with various configurations (top: 100% training data, bottom: 1% training data)

| Method | Stochastic Forest Fire ($p_{heat} = 0.90$) | Stochastic Host-Pathogen ($p_{cure} = 0.15$) | Stochastic Stock Market ($p_{invest} = 0.95$) |
|---|---|---|---|
| PredRNN (Wang et al., 2017) | $0.7984 \pm 0.0875$ | $0.5844 \pm 0.1488$ | $0.3991 \pm 0.1682$ |
| PredRNN++ (Wang et al., 2018) | $0.8081 \pm 0.0936$ | $0.5338 \pm 0.1608$ | $0.3919 \pm 0.1585$ |
| ImaGINator (Wang et al., 2020) | $0.4865 \pm 0.2000$ | $0.1774 \pm 0.1459$ | $0.2632 \pm 0.1183$ |
| SimVP (Gao et al., 2022) | $0.8441 \pm 0.0658$ | $0.6777 \pm 0.0974$ | $0.4050 \pm 0.1491$ |
| **RA-NCA** (Ours) | $\mathbf{0.8987 \pm 0.0598}$ | $\mathbf{0.7109 \pm 0.1065}$ | $\mathbf{0.6158 \pm 0.1745}$ |
| Method | Stochastic Forest Fire ($p_{heat} = 0.90$) | Stochastic Host-Pathogen ($p_{cure} = 0.15$) | Stochastic Stock Market ($p_{invest} = 0.95$) |
| PredRNN (Wang et al., 2017) | $0.4564 \pm 0.2384$ | $0.0375 \pm 0.1132$ | $0.1274 \pm 0.0753$ |
| PredRNN++ (Wang et al., 2018) | $0.4459 \pm 0.2069$ | $0.0256 \pm 0.0627$ | $0.1104 \pm 0.0488$ |
| ImaGINator (Wang et al., 2020) | $0.0005 \pm 0.0019$ | $0.2404 \pm 0.1415$ | $0.1611 \pm 0.0586$ |
| SimVP (Gao et al., 2022) | $0.7387 \pm 0.0992$ | $0.1576 \pm 0.1177$ | $0.2192 \pm 0.1105$ |
| **RA-NCA** (Ours) | $\mathbf{0.8672 \pm 0.0729}$ | $\mathbf{0.6957 \pm 0.1078}$ | $\mathbf{0.5762 \pm 0.1887}$ |

Table 15: Comparison of real-world wildfire prediction models. *(Huot et al., 2022) provides the results of the three classical methods.

| Method | AUC (PR) (%) ↑ |
|---|---|
| RA-NCA (Ours) | **29.2** |
| CNN-based (Huot et al., 2022) | 28.4 |
| *Random Forest | 22.5 |
| *Logistic Regression | 19.8 |
| *Persistence | 11.5 |

# E  REAL-WORLD APPLICATION: WILDFIRE PREDICTION

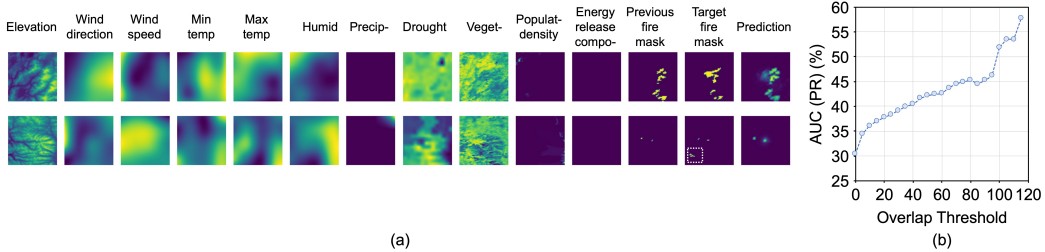

(a)                                                                                     (b)

Figure 19: Prediction results of RA-NCA in the real-world wildfire dataset (Huot et al., 2022). (a) shows the input data (∼12-th columns), target (13-th column), and prediction (14-th column). (b) shows the AUC (PR) by ignoring the test samples which exhibit fire ignition at random locations.

We investigate the performance of RA-NCA using a recently published real-world wildfire dataset (Huot et al., 2022). The dataset involves 64x64km historical wildfire since 2000 provided by NASA LPDAAC at the USGS EROS Center (data source from GEE (Gorelick et al., 2017)).

It mainly aims to predict the binary fire map (1: fire and 0: no fire) of next day given multivariate and continuous two-dimensional input, including topography, yesterday's fire map, vegetation, weather, etc. Huot et al. (2022) developed a convolutional neural network (not NCA-based) and evaluated with other methods, such as Random Forest, Logistic Regression, and Persistence (predicts today will be same as yesterday). The evaluation metric is AUC (PR) where the target output is a binary map (1: fire, 0: no fire). We evaluate the RA-NCA in the same test dataset. The comparison results are provided in Table 15.

RA-NCA exhibits better performance than the presented methods in the paper. We observe that RA-NCA provides reasonable prediction results particularly when the fire is locally diffused (Figure 19(a) top row). The failure scenarios are primarily observed when the next-day fire is ignited from completely new locations (Figure 19(a) bottom row), which is extremely challenging to predict. Since our model aims to model the local interaction, we also evaluate the performance by removing such test samples. We define Overlap Threshold to ignore test samples if the number of overlapped pixels between yesterday's fire mask and next day's fire mask do not exceed this value. That is, higher Overlap Threshold will ignore randomly ignited test cases and incorporate more portion of locally diffusing test cases. Figure 19(b) shows that, the AUC (PR) proportinally increase with the Overlap Threshold, indicating that RA-NCA performs much better on locally diffusing fire evolution. In summary, despite the non-ideal factors, RA-NCA shows the promising accuracy in the real-world wildfire dataset.

