# OpenReview forum: "Recurrent Neural Cellular Automata with Self-Attention for Multi-agent System"
_ICLR.cc/2024/Conference — ICLR 2024 Conference Withdrawn Submission_

### Official Review · Reviewer_kzFw · 2023-10-30

**Soundness:** 3 good
**Presentation:** 2 fair
**Contribution:** 3 good
**Rating:** 6
**Confidence:** 3

**Summary:**

The paper proposes a novel neural network model called Recurrent Attention-based Neural Cellular Automata (RA-NCA) for predicting the dynamics of many-agent systems with local and stochastic interactions, which has not been explored in existing work. The empirical results on the three synthetic cases show that the proposed method presents superior data efficiency and scalability.

**Strengths:**

- It addresses a challenging and underexplored problem of learning the hidden interaction rules in many-agent systems without prior knowledge or assumptions.
- It introduces a recurrent cellular attention module that combines LSTM and cellular self-attention to capture the temporal and permutation-invariant information of neighboring agents.
- It demonstrates the superior performance of RA-NCA over other NCA networks and video prediction networks in terms of data efficiency, robustness, and scalability across three synthetic datasets with different levels of stochasticity.
- The experimental results are conducted thoroughly and convincingly.

**Weaknesses:**

- It lacks an investigation on the scale of considered neighbors. Specifically, the authors consider only the case of Moore's neighborhood.
- It only evaluates RA-NCA on synthetic datasets, which may not reflect the complexity and diversity of real-world many-agent systems.
- It does not compare RA-NCA with other state-of-the-art methods for multi-agent interaction learning, e.g., networked agent learning derived from distributed optimization [1], which leverages a similar philosophy to resolve large-scale agent learning. Nonetheless, I do not regard it as a reason to reject this paper, and I hope the authors give further comparison in this paper, experimentally or conceptually.
- Some key concepts lack clarity, such as what is agent interaction? I suggest the authors give a brief introduction in the "Related work and Background".



[1] Zhang, K., Yang, Z., Liu, H., Zhang, T., & Basar, T. (2018, July). Fully decentralized multi-agent reinforcement learning with networked agents. In International Conference on Machine Learning (pp. 5872-5881). PMLR.

**Questions:**

1. Section 2: "However, recurrent neural networks based approaches have been ....", I didn't get the point of this claim.
2. Section 2: "However, these methods are not explicitly designed to preserve spatial structures in latent space ...". The question is that why can NCA preserve spatial structures in latent space?
3. Section 3: "Recently, graph-based, attention-based ,...". Please add references to existing work.
4. How does the NCA make the global interactions keep consistent with local interactions in Moore's neighborhood?
5. What is $c_{(i,j)}$ in equation (4)?
6. What is the value of $(t_{n_{pred}} - t_{n_{obs}})$ used for experiments?
7. Have the authors ever tried other training scales? expect for $64\times64$.

---

> ### Author Response · Authors · 2023-11-23
> **Response to the Reviewer kzFw 1/4**
>
> We thank the reviewer for their insightful comments and constructive feedback on our manuscript. We appreciate the opportunity to clarify and expand upon key aspects of our work. We hope we have addressed their concerns and questions regarding the paper and hope they will reconsider their rating.
>
> > Some key concepts lack clarity, such as what is agent interaction? I suggest the authors give a brief introduction in the "Related work and Background".
>
> **Reply**: Thank you for the suggestion. In the current manuscript, we clearly state that “Multi-agent systems in this paper specifically refers a system where stationary pixel agents with discrete states are locally interacting with their neighboring agents. The agent interaction is defined by the pre-defined rules that change the state of agents.” In the section 2 (Related Work and Background – Dataset: Synthetic Multi-agent System) Also, the details of interaction rules in the host-pathogen dataset is further described.
>
> > It does not compare RA-NCA with other state-of-the-art methods for multi-agent interaction learning, e.g., networked agent learning derived from distributed optimization [1], which leverages a similar philosophy to resolve large-scale agent learning. Nonetheless, I do not regard it as a reason to reject this paper, and I hope the authors give further comparison in this paper, experimentally or conceptually.
>
> **Reply**: A. Thank you for the important feedback. We agree that the reference paper also had a similar philosophy; for example, the authors emphasize “fully-decentralized environments”, where each agent is only allowed to access local information and controlled by a local controller [1].
>
> However, this work is generally focused on the control problem, not for the prediction, as the authors said “This framework of networked multi-agent systems finds a broad range of applications in distributed cooperative control problems.” (page 7 [1]). While there are few works trying to employ reinforcement learning (RL) in sequence modeling problems [2,3], the general challenge in RL is the low sample efficiency [4], which might be particularly critical in large stochastic systems with long sequences like our environments. As our primary goal in this paper is to demonstrate the efficacy of RA-NCA in extremely data-limited environments, we mainly study the recurrent neural networks-based predictive modeling in this paper.
>
> In summary, we agree that more discussion on these approaches should be included in the paper. We add the discussion on the RL-based methods in the section 2 (Related Work and Background - Interaction Learning in Multi-agent System). Additionally, we involve more experimental and conceptual comparisons with the state-of-the-art NCA model (non-RL based) [4] in Appendix C (Challenges in Employing prior NCA networks).
>
> [1] Zhang, K., Yang, Z., Liu, H., Zhang, T., & Basar, T. (2018, July). Fully decentralized multi-agent reinforcement learning with networked agents. In International Conference on Machine Learning (pp. 5872-5881). PMLR.
>
> [2] Janner, Michael, Qiyang Li, and Sergey Levine. "Offline reinforcement learning as one big sequence modeling problem." Advances in neural information processing systems 34 (2021): 1273-1286.
>
> [3] Ganapathi Subramanian, Sriram, and Mark Crowley. "Using spatial reinforcement learning to build forest wildfire dynamics models from satellite images." Frontiers in ICT 5 (2018): 6.
>
> [4] Micheli, Vincent, Eloi Alonso, and François Fleuret. "Transformers are sample efficient world models." arXiv preprint arXiv:2209.00588 (2022).
>
> [5] Tesfaldet, Mattie, Derek Nowrouzezahrai, and Chris Pal. "Attention-based Neural Cellular Automata." Advances in Neural Information Processing Systems 35 (2022): 8174-8186.

---

> ### Author Response · Authors · 2023-11-23
> **Response to the Reviewer kzFw 2/4**
>
> > It lacks an investigation on the scale of considered neighbors. Specifically, the authors consider only the case of Moore's neighborhood.
>
> **Reply**: We also investigate another scale of 5x5 neighborhood in RA-NCA. The training and evaluation environment is the deterministic forest fire model with the full amount of data. The interaction rules in the dataset remain the same. It is important to mention that the 5x5 model does not know about the interaction scale in the forest fire. We first compare the training dynamics (training and validation BCE loss) in the RA-NCA (3x3 neighborhood) and RA-NCA (5x5 neighborhood) models (Figure 11 in Appendix B). The training speed is different in the two models, but their minimum validation loss is converged into the similar level. Also, in terms of F1 and AUC scores, the performance of the two models are similar (F1 is higher in 3x3 but AUC is higher in 5x5). The results are given as follows:
>
> | Model | F1 | AUC |
> |--------|--------|--------|
> | RA-NCA (3x3) | 0.8880±0.0610 | 0.9766±0.0219 |
> | RA-NCA (5x5) | 0.8786±0.0592 | 0.9787±0.0195 |
>
> We include this discussion and experimental results in Appendix B (Different Neighborhood Size in RA-NCA) (see Table 6). However, we would like to also mention that 3x3 Moore’s neighborhood is the most common configuration in Cellular Automata, and hence neural CA architectures are generally aimed to perform on the 3x3 Moore’s neighborhood [1-4].
>
> [1] Mordvintsev, Alexander, et al. "Growing neural cellular automata." Distill 5.2 (2020): e23.
>
> [2] Gilpin, William. "Cellular automata as convolutional neural networks." Physical Review E 100.3 (2019): 032402.
>
> [3] Hernandez, Alejandro, Armand Vilalta, and Francesc Moreno-Noguer. "Neural cellular automata manifold." Proceedings of the IEEE/CVF Conference on Computer Vision and Pattern Recognition. 2021.
>
> [4] Earle, Sam, et al. "Illuminating diverse neural cellular automata for level generation." Proceedings of the Genetic and Evolutionary Computation Conference. 2022.
>
>
> > It only evaluates RA-NCA on synthetic datasets, which may not reflect the complexity and diversity of real-world many-agent systems.
>
> **Reply**: We agree that it is important to evaluate the RA-NCA on the real-world many-agent systems. However, we were unfortunately unable to find common benchmarks, which might be because the problem itself is relatively novel and unexplored. Instead, we investigate the efficacy of RA-NCA using a real-world wildfire dataset [1]. We incorporate the following discussion and experimental results in Appendix E and also mention it as a real-world application of RA-NCA in the section 2 (Related Work and Background – Dataset: Synthetic Multi-agent System)
>
> This dataset includes 64x64km historical wildfire since 2000 provided by NASA LPDAAC at the USGS EROS Center (data source from GEE [2]). The main objective in this dataset is to predict the binary fire map (1: fire and 0: no fire) of next day using multivariate (12 channels) two-dimensional input. It includes yesterday’s fire map, weather, vegetation, etc. As the authors [1] provide the benchmark performances employing their CNN-based model, some classical methods (Random Forest, Logistic Regression), and persistence (next day is same as yesterday), we directly compare RA-NCA’s accuracy with these values. The comparison table is provided below (*the results from [1]):
>
> | Model      | AUC (PR) (%) |
> | ----------- | ----------- |
> | RA-NCA      | 29.2 |
> | CNN-based [1]  | 28.4 |
> | *Random Forest | 22.5 |
> | *Logistic Regression | 19.8 |
> | *Persistence | 11.5 |
>
> The evaluation is based on AUC (PR) from their paper, where the positive value corresponds fire (otherwise, no fire). The table showcases that, RA-NCA provides better predictions thant the presented methods in the paper. In particular, RA-NCA generates reasonable prediction results when the fire is locally diffused (Figure 19(a) top row). The failure scenarios are primarily observed when the next-day fire is ignited from completely new locations (Figure 19(b) bottom row), which is even not diffused from yesterday’s fire.

---

> ### Author Response · Authors · 2023-11-23
> **Response to the Reviewer kzFw 3/4**
>
> In this light, we further characterize the performance of RA-NCA using Overlap Threshold. We define Overlap Threshold to ignore test samples if the number of overlapped fire pixels between yesterday's fire mask and today's fire mask does not exceed this value. For example, if Overlap Threshold is 0, we do not consider the test samples where yesterday's fire mask and next day's fire mask are not overlapped at all (randomly ignited). Higher Overlap Threshold will ignore more randomly ignited test samples. The accuracy considering the Overlap Threshold is given below:
>
> | Overlap Threshold      | AUC (PR) (%) |
> | ----------- | ----------- |
> | 0     | 30.4 |
> | 10 | 36.1 |
> | 20 | 37.7 |
> | 30 | 39.2 |
> | 40 | 40.4 |
> | 50 | 42.2 |
> | 60 | 42.6 |
> | 70 | 44.5 |
> | 80 | 45.3 |
> | 90 | 45.2 |
> | 100 | 51.8 |
> | 110 | 53.5 |
>
> The key observation is that, the AUC (PR) of RA-NCA gradually increases with the higher Overlap Threshold. That is, the performance of RA-NCA in the locally evolving wildefire is much better. In summary, RA-NCA exhibits the promising performance in the real-world example as well despite the non-ideal factors.
>
> [1] Huot, Fantine, et al. "Next day wildfire spread: A machine learning dataset to predict wildfire spreading from remote-sensing data." IEEE Transactions on Geoscience and Remote Sensing 60 (2022): 1-13.
>
> [2] N. Gorelick, M. Hancher, M. Dixon, S. Ilyushchenko, D. Thau, and R. Moore, “Google Earth engine: Planetary-scale geospatial analysis for everyone,” Remote Sens. Environ., vol. 202, pp. 18–27, Dec. 2017
>
>
> > Have the authors ever tried other training scales? expect for 64x64.
>
> **Reply**: Thank you for the interesting question. We have investigated two other training scales, 32x32 and 48x48. We train the models in the total three scales (32, 48, and 64) and evaluate in the same large scale (512). The training environments involve stochastic forest fire (p_{heat}=0.90), host-pathogen (p_{cure}=0.15), and stock market (p_{invest}=0.95) with full amount of data. The results are provided below:
>
> | Model (train->test) | Forest Fire | Forest Fire    | Host-Pathogen |  Host-Pathogen   | Stock Market | Stock Market    |
> |---------------------|:----------------------:|:---:|:------------------------:|:---:|:-----------------------:|:---:|
> |                                   |           F1            | AUC                   |            F1            | AUC |            F1           | AUC |
> | RA-NCA (64->512)    |  0.9172±0.0196 | 0.9878±0.0079  | 0.7241±0.0403 | 0.9593±0.0220 | 0.6177±0.1467 | 0.7758±0.0706  |
> | RA-NCA (48->512)    |  0.9163±0.0187 | 0.9872±0.0085  | 0.7196±0.0427 | 0.9585±0.0225 | 0.6175±0.1467 | 0.7758±0.0705  |
> | RA-NCA (32->512)    |  0.9114±0.0215 | 0.9852±0.0088  | 0.7161±0.0422 | 0.9579±0.0227 | 0.6112±0.1501 | 0.7756±0.0705  |
>
> We observe that the models perform better in the higher scales, but their performance disparity is not noticeable. This results provide an important insight in learning the locally-interacting systems, since reducing the training scale from 512 to 32 can significantly save the training computational costs.
>
>
> To further illustrate the effect of training scales, we explore the data-limited setting as well. The table below shows the results trained with 1% of training data:
>
> | Model (train->test) | Forest Fire | Forest Fire    | Host-Pathogen |  Host-Pathogen   | Stock Market | Stock Market    |
> |---------------------|:----------------------:|:---:|:------------------------:|:---:|:-----------------------:|:---:|
> |                                   |           F1            | AUC                   |            F1            | AUC |            F1           | AUC |
> | RA-NCA (64->512)    |0.8921±0.0194|0.9778±0.0109|0.7061±0.0446|0.9463±0.0275|0.5775±0.1633|0.7728±0.0714|
> | RA-NCA (48->512)    |0.8782±0.0228|0.9728±0.0127|0.6746±0.0531|0.9215±0.0269|0.5422±0.1742|0.7704±0.0729|
> | RA-NCA (32->512)    |0.8851±0.0217|0.9635±0.0132|0.6262±0.0595|0.9239±0.0395|0.5660±0.1617|0.7693±0.0728|
>
> An interesting observation is that the performance degradation resulted from the limited training data is more noticeable in lower scales. For example, in Host-Pathogen, the F1 score of 32->512 case drops by 0.089 by reducing the training data 100->1%, while that of 64->512 case drops only by 0.018. This indicates the correlation between the data efficiency and system scales. Our understanding is that there will be more chances for the model to observe agent interactions in higher scales. For example, 1% training data of 64x64 will involve approximately 4 times more agent interactions than 1% training data of 32x32. Thank you for suggesting this experiment. We discuss this observation in the new paragraph of section 4.1 (Comparison with other NCA Networks - Scalability: Evaluation in Large Stochastic Systems).

---

> > ### Author Response · Authors · 2023-11-23
> > **Response to the Reviewer kzFw 4/4**
> >
> > > Section 2: "However, these methods are not explicitly designed to preserve spatial structures in latent space ...". The question is that why can NCA preserve spatial structures in latent space?
> >
> > **Reply**: NCA architectures are generally focused on performing on the individual cell, by calculating the update for each of cells, using the 3x3 neighboring information. For example, if input dimension is R^{HxWxC}, (H, W: height and width of input, C: channel), then latent space is R^{HXWXC’}, and final output is R^{HxWxC} or R^{HxWx1} (in our case). In other words, the models perform agent-wise processing, which only changes the channel dimension (C->C’->C or 1) preserving the spatial dimension (HxW -> HxW -> HxW). We add the discussion in Appendix D (Comparison with Video Prediction Networks – Unphysical Prediction in Video Prediction Networks)
> >
> > > Section 2: "However, recurrent neural networks based approaches have been ....", I didn't get the point of this claim.
> >
> > **Reply**: Our claim was that the prior NCA architectures have rarely employed recurrent neural networks since their applications are mostly focused on stationary data. In order to avoid the confusion, we re-write the section 2 (Related Work and Background - Neural Cellular Automata).
> >
> > > Section 3: "Recently, graph-based, attention-based ,...". Please add references to existing work.
> >
> > **Reply**: Thank you for the comment. We added references [1-3] in the section 3 (Proposed Approach).
> >
> > [1] Grattarola, Daniele, Lorenzo Livi, and Cesare Alippi. "Learning graph cellular automata." Advances in Neural Information Processing Systems 34 (2021): 20983-20994.
> >
> > [2] Tesfaldet, Mattie, Derek Nowrouzezahrai, and Chris Pal. "Attention-based Neural Cellular Automata." Advances in Neural Information Processing Systems 35 (2022): 8174-8186.
> >
> > [3] Palm, Rasmus Berg, et al. "Variational neural cellular automata." arXiv preprint arXiv:2201.12360 (2022).
> >
> > > How does the NCA make the global interactions keep consistent with local interactions in Moore's neighborhood?
> >
> > **Reply**: Thanks for the interesting question. In our problems, we assume that the system evolution is purely governed by the local interactions. Thus, our claim is primarily in learning the local interactions and does not assume that NCA can effectively learn the global interactions.
> >
> > Since NCA can only operate on the Moore's neighborhood, there is no direct (and fast) influence outside from the Moore's neighborhood. However, when CA processes 3x3 areas for each of agents (similar to 3x3 convolution with stride 1), there exists an overlap in the processing areas (similar to overlapped sliding window in convolution), and it can slowly deliver the local information to the global areas [1,2]. Investigating the global interactions in the NCA will be an interesting future direction.
> >
> > [1] Phipps, Michel J. "From local to global: the lesson of cellular automata." Individual-based models and approaches in ecology. Chapman and Hall/CRC, 2018. 165-187.
> >
> > [2] Jen, Erica. "Global properties of cellular automata." Journal of Statistical Physics 43 (1986): 219-242.
> >
> > > What is c_{(i,j)} in equation (4)?
> >
> > **Reply**: We apologize for the confusion. It is a cell state (c) in LSTM for an agent at (i, j) (i.e., i-th row and j-th col in the system). We add the description for the model details in the section 3  (Proposed Approach).
> >
> > > What is the value of t_n_{pred} - t_n_{obs} used for experiments?
> >
> > **Reply**: We set t_n_{obs}=10 for the three datasets in common and t_n_{pred}=60 for the forest fire model and t_n_{pred}=30 for the host-pathogen and stock market models. We should have clarified the values in the experiment section. We specify the values in the section 4 (Experimental Result - Experimental Setting).

---

### Official Review · Reviewer_jsLT · 2023-10-30

**Soundness:** 3 good
**Presentation:** 3 good
**Contribution:** 3 good
**Rating:** 3
**Confidence:** 4

**Summary:**

The paper proposes a new model for many-agent system prediction/simulation such as epidemic spread, rumor propagation through crowd,
prey-predator model, and forest fire. Unlike the previous CNN-based model which suffers from spatial dependency, the proposed new model
 is based on self-attention which is permutation invariant and thus overcomes this issue. In addition, the LSTM is introduced to endow the model with memory, which is essential to leverage historical information to do prediction. The experiment section proves the effectiveness of the method and shows that the model outperforms other baselines even in extremely data-limited scenarios.

**Strengths:**

1. The paper is well-written and easy to understand
2. The motivation for model designing makes sense
3. The result shows it outperforms baselines in this field and other video prediction baselines

**Weaknesses:**

1. The method is trained and evaluated on in-house image datasets collected with stale simulators (released 20 years ago). As I am not an expert in the related field, I wonder if there are some new and common benchmarks for evaluating the method like real-world forest fire data, which should be unstructured and sophisticated.
2. I can not find any potential for applying the method to important problems. The method represents the many-agent system with a structured semantic image and converts the problem to an image prediction problem on toy datasets.
3. The method is a simple combination of two existing and well-studied techniques. Thus technical novelty is limited as well.

**Questions:**

N/A

---

> ### Author Response · Authors · 2023-11-23
> **Response to the Reviewer jsLT 1/2**
>
> We sincerely thank the reviewer for your time to review the paper. We hope we have addressed their concerns and questions regarding the paper and hope they will reconsider their rating.
>
> > The method is a simple combination of two existing and well-studied techniques. Thus technical novelty is limited as well.
>
> **Reply**: Thank you for the feedback. We should have clearly mentioned our contributions and why these are technically novel. Our RA-NCA is a novel NCA architecture featuring memory function and permutation invariance. While prior works on NCAs have not explored the memory function and permutation invariance together, we understand that recurrent neural networks and self-attention are existing techniques. However, our key technical novelty is in “identifying the benefits of the novel NCA-based architecture, primarily data-efficiency and scalability, in locally interacting multi-agent systems.”
>
> This provides an important yet underexplored research insight in the field of NCAs since other non-NCA methods generally require a large amount of training data to discover the hidden interaction rules and need to re-train their models depending on the system scale. Reviewer o3LN also stated “regarding the proposed architecture, both key contributions of showcasing the data-efficiency and the model-scalability pose an interesting topic of research”. Accordingly, our method empirically demonstrates the efficacy of RA-NCA in the extremely data-limited scenario (1%) and 36x (train 64x64 -> test 512x512) larger systems. In summary, we believe that our technical novelty is not limited and will rather open up interesting future research in the NCA domain. We additionally clarify our technical novelty in the introduction.
>
>
> > I can not find any potential for applying the method to important problems. The method represents the many-agent system with a structured semantic image and converts the problem to an image prediction problem on toy datasets.
>
> **Reply**: We understand the reviewer’s concern. The systems we are trying to address are driven by many structured dynamical systems (referred to as “agents”) with local and stochastic interactions. These are the fundamental characteristics in the wide range of important problems, such as not only forest fire but also epidemic spread or rumor propagation in crowds. However, our goal is not to develop a directly applicable solution for one specific real-world problems, rather understand the critical bottlenecks in learning the interaction in the non-ideal environments (e.g., stochastic or/and extremely limited data) when deploying existing NCA-based and scene-based prediction networks; and explore methods (like the RA-NCA) to address them.
>
> In this regard, we believe the synthetic datasets are particularly useful for in-depth understanding of the model performance since we can generate the various scenarios including different stochasticity levels and system scales. While the task can be translated into an image prediction problem, it is still not a conventional image prediction problem since pixel-level dynamics and their correlation are crucial. Even if the systems seem to be relatively simple, the scene/video prediction networks, as the reviewer mentioned in "Strengths 3", do not perform well in these systems with a huge accuracy disparity compared to RA-NCA. In this light, our RA-NCA demonstrates the two contributions, which can be applied to the wide range of important problems: 1) a novel NCA solution for discovering hidden interactions rules in locally interacting multi-agent systems and 2) another perspective to process this type (governed by local interaction) of sequential images by treating each of pixels as locally interacting agents. In the current manuscript, we explicitly state the second contribution in the section 4.2 (comparison with video prediction networks).

---

> ### Author Response · Authors · 2023-11-23
> **Response to the Reviewer jsLT 2/2**
>
> > I wonder if there are some new and common benchmarks for evaluating the method like real-world forest fire data, which should be unstructured and sophisticated.
>
> **Reply**: Thank you for the suggestion. We also consider the evaluation of our model on the real-world scenario as an important research direction. It was unfortunately difficult to find common benchmarks given the problem itself is relatively new and underexplored. However, as the reviewer suggested, we investigate the performance of RA-NCA using a recently published real-world wildfire dataset [1]. The dataset involves 64x64km historical wildfire since 2000 provided by NASA LPDAAC at the USGS EROS Center (data source from GEE [2]).
>
> It mainly aims to predict the binary fire map (1: fire and 0: no fire) of next day given multivariate and continuous two-dimensional input, including topography, today’s fire map, vegetation, weather, etc. The authors in the paper developed a convolutional neural network (not NCA-based) and evaluated with other methods, such as Random Forest, Logistic Regression, and Persistence (predicts tomorrow will be same as today). The evaluation metric is AUC (PR) where the target output is a binary map (1: fire, 0: no fire). We train and evaluate the RA-NCA in the same dataset. The comparison results are provided in the table below (*the accuracy of these methods is also provided in [1]):
>
> | Model      | AUC (PR) (%) |
> | ----------- | ----------- |
> | RA-NCA      | 29.2 |
> | CNN-based [1]  | 28.4 |
> | *Random Forest | 22.5 |
> | *Logistic Regression | 19.8 |
> | *Persistence | 11.5 |
>
> RA-NCA exhibits better performance than the presented methods in the paper. We observe that RA-NCA provides reasonable prediction results particularly when the fire is locally diffused (Figure 19(a) top row). The failure scenarios are primarily observed when the next-day fire is ignited from completely new locations (Figure 19(b) bottom row), which is not diffused from yesterday’s fire and hence extremely challenging to predict. Since our model aims to model the local interaction, we also evaluate the performance by removing such non-local test samples. We define Overlap Threshold to ignore test samples if the number of overlapped fire pixels between yesterday's fire mask and today's fire mask does not exceed this value. For example, if Overlap Threshold is 0, we do not consider the test samples where yesterday's fire mask and next day's fire mask are not overlapped at all (randomly ignited). That is, higher Overlap Threshold will ignore randomly ignited test cases and incorporate more portion of locally diffusing test cases. The table below showcases that AUC (PR) of RA-NCA gradually increases with the higher Overlap Threshold, indicating that RA-NCA actually performs much better on the locally diffusing fire evolution. In summary, despite the non-ideal factors observed in the dataset, RA-NCA exhibits the promising accuracy in the real-world forest fire dataset than the prior work. We mention the real-world application of RA-NCA in section 2 (Related Work and Background - Data: Synthetic Multi-agent System) and include this discussion and results in Appendix E.
>
> | Overlap Threshold      | AUC (PR) (%) |
> | ----------- | ----------- |
> | 0     | 30.4 |
> | 10 | 36.1 |
> | 20 | 37.7 |
> | 30 | 39.2 |
> | 40 | 40.4 |
> | 50 | 42.2 |
> | 60 | 42.6 |
> | 70 | 44.5 |
> | 80 | 45.3 |
> | 90 | 45.2 |
> | 100 | 51.8 |
> | 110 | 53.5 |
>
> [1] Huot, Fantine, et al. "Next day wildfire spread: A machine learning dataset to predict wildfire spreading from remote-sensing data." IEEE Transactions on Geoscience and Remote Sensing 60 (2022): 1-13.
>
> [2] N. Gorelick, M. Hancher, M. Dixon, S. Ilyushchenko, D. Thau, and R. Moore, “Google Earth engine: Planetary-scale geospatial analysis for everyone,” Remote Sens. Environ., vol. 202, pp. 18–27, Dec. 2017

---

### Official Review · Reviewer_o3LN · 2023-10-31

**Soundness:** 2 fair
**Presentation:** 3 good
**Contribution:** 1 poor
**Rating:** 3
**Confidence:** 3

**Summary:**

This paper introduces Recurrent Attention-based Neural Cellular Automata (RA-NCA), combining the attention mechanic with the recurrent LSTM neural network for state prediction in cellular automata (CA). In contrast to other CA approaches using CNNs as their feature projector, the proposed architecture  is presented as superior regarding training and sample efficiency, mainly due to the attention’s permutation invariance. RA-NCA is also shown to scale well in the transfer to larger CA systems. For the evaluation, three CA settings are being tested (forest fire, host-pathogen spread and a stock-marked model), comparing RA-NCA to a LSTM+CNN model and an ablation with only the attention mechanism. RA-NCA was also evaluated with respect to sample efficiency and compared to recent Scene/Video-Prediction Networks on three levels of stochastic transition for the forest fire domain.

**Strengths:**

The paper is concisely written and understandable. All key-concepts are explained and cleanly formalized. The domains are well described in the Appendix. The evaluation on three domains covers diverse aspects with regards to density and the amount of training data. The focus point of the attention invariance is well explained and nicely visualized in Fig. 2 and 3. Nevertheless, I would like to point in the direction of Tang and Ha (2021), which is quite close in terms of insights, but applied for the field of RL. A discussion on the key difference to this work, perhaps with focus on the (Moore) neighborhood, would certainly benefit the quality of this work. Regarding the proposed architecture, both key contributions of showcasing the data-efficiency and the model-scalability pose an interesting topic of research, but I would have liked to see some more focus on the latter.

Y. Tang and D. Ha, "The Sensory Neuron as a Transformer: Permutation-Invariant Neural Networks for Reinforcement Learning", in Advances in Neural Information Processing Systems, 2021.

**Weaknesses:**

While the paper contains some interesting foundations, it lacks focus on the field of CAs by mixing in notions of multi-agent systems and video prediction networks. From my view of (MA)RL and Game Theory, the mention of CA as multi-agent systems seems quite unintuitive, since I understand CAs more as conditional transition (probabilities) between cells rather than multiple agents interacting with each other. Apart from the notation and the section on Multi-Agent interaction, I also see no distinction being made between classical neural CAs and each cell being an agent.

Also, the evaluation could be improved. Evaluating only against two variation baselines (CNN+LSTM, only Self-Attention) seems to be on the weaker side. Convolutional-recurrent Networks are being steadily phased out of image recognition in favor of e.g., Vision-Transformers, in part due to sample efficiency of training convolutional kernels. The claim “The three networks consist of the similar level of trainable parameters.” on page 7 is similarly questionable, since CNN+LSTM and RA-NCA almost double the parameter count of Attention-CA. Since taking the hidden-state information out of the Attention-CA baseline seems to be impactful, comparing to Attention+RNN would perhaps be more insightful.
I would have also liked to see a comparison to the obvious state-of-the-art architecture Attention-Transformer NCA  (Tesfaldet et al., 2022), that is cited but not used in the comparison between Transformer and LSTM. Even if this comparison is intended as part of a pre-study on the importance of the respective architecture components, there is still a significantly overlap to the section on video prediction networks.

The inclusion of the comparison to video prediction networks seems to overall weaken the focus of the paper without providing novel insights. Since I am not too familiar with video prediction networks I cannot speak to the quality of the chosen models as baselines. However, since the CA-video is reduced to minimal color diversity that simply represent very few states (binary even, as it is evaluated / trained), apart from the concept of "predicting batches of pixels from previous batches of pixels", I see not much of a connection to the field of NCAs. Furthermore, CNN+LSTM (and simple Attention) as Model Representatives are already covered in the NCA study and are already evaluated on the same settings as Fig. 6 and Fig. 7. The space dedicated to both the multi-agent interaction and video prediction networks feels disconnected from the core-idea and could have been used to round out the otherwise very interesting component-study in the main-field of NCAs for a more focused and in-depth paper. Also, the discussion on original cell-states on page 14 in the Appendix could perhaps deserve a mention in the main paper?

I would also recommend to more fairly mention the advantages and limitations (e.g., parameter disparity of the Attention-CA) of the other baselines and expand on the discussion. Comparable (or better) performance of the CNN+LSTM in Fig.5 (forest fire) up until data-amount of around 10% could be more discussed, as could
the good results on the other two domains (that are only found in Table 5 in the Appendix). I would also like to see the std-div. in the main paper tables (similar to the ones in the Appendix) and some more discussion on variance, since e.g. Fig.6 (the zoomed out bar-plots) shows how close performance is if you consider the variance.

In summary, I would recommend to shift the focus away from multi-agent interactions and video-prediction networks to focus and elaborate more on the two core-insights of this paper (Low-Data Efficiency and Scalability) to improve focus and balance of the paper. The scaling aspect in particular could offer an interesting insight into the degrees of scalability. It would be encouraged to show a direct comparison to the Transformer-NCA and make the Attention-CA baseline more comparable in terms of parameters to gain a better overview on the current state of the field of NCAs in
general.

**Questions:**

Could you please specify on the $\bigodot$ operator in Eq. 4?
Why was 32 chosen as the encoding dimension?
Is there a reason why the Transformer Architecture is not applicable in this setting? Would you consider using the Attention-Transformer NCA from Tesfaldet et al., 2022 as a more competitive baseline?
Could you please motivate/elaborate why you are considering NCA-cells as interacting agents. Apart from the summation of the neighbor’s hidden state there is not “interaction” in the classical multi-agent sense, that I would have understood here?

Minor Comments:
p3: “Recently, graph-based, attention- based, variational autoencoder-based NCA networks are also proposed.” could use citations, as could the “classical” ML-LSTM literature (e.g. Schmidthuber et al) that you are building on.

---

> ### Author Response · Authors · 2023-11-23
> **Response to the Reviewer o3LN 1/7**
>
> We are glad to know that the reviewer considers our work interesting. We sincerely thank the reviewer for their constructive feedback. We hope we have addressed their concerns and questions regarding the paper and hope they will reconsider their rating.
>
> > From my view of (MA)RL and Game Theory, the mention of CA as multi-agent systems seems quite unintuitive, since I understand CAs more as conditional transition (probabilities) between cells rather than multiple agents interacting with each other.
>
> > Could you please motivate/elaborate why you are considering NCA-cells as interacting agents. Apart from the summation of the neighbor’s hidden state there is not “interaction” in the classical multi-agent sense, that I would have understood here?
>
> **Reply**: Thank you for the insightful feedback. We understand with the reviewer’s viewpoint that multi-agent systems are generally referred in RL domains and involve “intelligent” agents that have a reasoning capability.
>
> In contrast, our cells follow pre-defined transition rules and not intelligent. However, CA has been employed to simulate such multi-agent systems with non-intelligent agents [1-3]. In this light, our system is also non-intelligent and interacting multi-agent systems. We would like to also note that, apart from CA, non-intelligent such as bouncing balls (Figure 3. in [4]) and particle physics system (Figure 2. in [5]) are also studied in the context of multi-agent systems. We believe that the confusion comes from that the unclear definition of multi-agent systems in the paper.
>
> To avoid the confusion, we clearly state the definition of multi-agent systems we are addressing: “Multi-agent systems in this paper specifically refers a system where stationary pixel agents with discrete states are locally interacting with their neighboring agents.” in the section 2 (Related Work and Background section - Dataset: Synthetic Multi-agent System paragraph).
>
> We also understand that it may be still unintuitive as other reviewers also provide the similar concern. As an alternative term for this system, we also consider “locally interacting discrete dynamical systems” instead of multi-agent system. This is inspired from the definition of CA: “discrete spatiotemporal dynamic systems based on local rules” [6].
>
> We would be happy to change the title accordingly in the final version (if accepted) if the reviewers suggests and conference policy allows that.
>
> [1] Wang, Jinghui, et al. "A multi-agent based cellular automata model for intersection traffic control simulation." Physica A: Statistical Mechanics and its Applications 584 (2021): 126356.
>
> [2] Liu, Yaolin, et al. "Simulating the conversion of rural settlements to town land based on multi-agent systems and cellular automata." PloS one 8.11 (2013): e79300.
>
> [3] Dijkstra, Jan, Joran Jessurun, and Harry JP Timmermans. "A multi-agent cellular automata model of pedestrian movement." Pedestrian and evacuation dynamics 173 (2001): 173-180.
>
> [4] Hoshen, Yedid. "Vain: Attentional multi-agent predictive modeling." Advances in neural information processing systems 30 (2017).
>
> [5] Li, Jiachen, et al. "Evolvegraph: Multi-agent trajectory prediction with dynamic relational reasoning." Advances in neural information processing systems 33 (2020): 19783-19794.
>
> [6] Clarke, Keith C. "Cellular automata and agent-based models." Handbook of regional science (2014): 1217-1233.

---

> ### Author Response · Authors · 2023-11-23
> **Response to the Reviewer o3LN 2/7**
>
> > The claim “The three networks consist of the similar level of trainable parameters.” on page 7 is similarly questionable, since CNN+LSTM and RA-NCA almost double the parameter count of Attention-CA.
>
> > make the Attention-CA baseline more comparable in terms of parameters
>
> > I would also recommend to more fairly mention the advantages and limitations (e.g., parameter disparity of the Attention-CA) of the other baselines and expand on the discussion.
>
> **Reply**: We agree with the reviewer’s feedback. We design a large Attention-CA baseline with 20,297 trainable parameters. Now, the number of parameters in the models is ConvLSTM-CA (21,169), RA-NCA (19,065), Attention-CA (small: 10,617, large: 20,297). We studied the performance of the Large Attention-CA in stochastic and data-limited settings. The training environments are stochastic forest fire (p_{heat}=0.9), host-pathogen (p_{cure}=0.15), stock market (p_{invest}=0.95) with the 1% amount of training data. As shown the table below, the Large Attention-CA shows poorer performance compared to our RA-NCA. In fact, we observe that, the training of Large Attention-CA is relatively unstable than the Small Attention-CA, and consequently showing lower F1 and AUC (ROC) scores in the three datasets.
>
> | Model (parameters) | Forest Fire | Forest Fire    | Host-Pathogen |  Host-Pathogen   | Stock Market | Stock Market    |
> |---------------------|:----------------------:|:---:|:------------------------:|:---:|:-----------------------:|:---:|
> |                                    |           F1            | AUC                   |            F1            | AUC |            F1           | AUC |
> |Attention-CA (10,617) | 0.8305±0.0822 | 0.9705±0.0283 | 0.6759±0.1159 | 0.9433±0.0372 | 0.5103±0.1862 | 0.7692±0.0829 |
> |Attention-CA (20,297) | 0.2880±0.1814 | 0.8118±0.1338 | 0.6770±0.1109 | 0.9403±0.0378 | 0.5050±0.1917 | 0.7684±0.0838 |
> |RA-NCA (19,065)        | 0.8672±0.0729 | 0.9768±0.0220 | 0.6957±0.1078 | 0.9443±0.0405 | 0.5762±0.1887 | 0.7751±0.0855 |
>
>
> This indicates that simply increasing the model complexity of Attention-CA does not help learning the interaction rules in stochastic and data-limited scenarios.
>
> In the later answers, we also explore our RA-NCA models with various sizes, and interestingly, RA-NCA with 6,185 parameters still outperforms Attention-CA in the Stock Market dataset. In summary, our RA-NCA still demonstrates higher F1 and AUC (ROC) scores than the Attention-CA with the similar level of parameters. We delete the claim ("The three networks consist of the similar level of trainable parameters.") and re-write it in the section 4.1 (Comparison with other NCA networks): “The results on larger Attention-CA with 20k parameters are provided in Appendix C.” (see Table 8)
>
> > Since taking the hidden-state information out of the Attention-CA baseline seems to be impactful, comparing to Attention+RNN would perhaps be more insightful.
>
> **Reply**: Thanks for the insightful suggestion. We investigate the Attention+RNN model in our RA-NCA architecture. Its design is basically similar with RA-NCA, but we replace the LSTM module (Figure 3(b) right) with the basic RNN module. The hidden state in the RNN module is defined by: h_{t} = tanh(W_{ih}x_{t} + b_{ih} + W_{hh}h_{t-1} + b_{hh}). The training environments are stochastic forest fire (p_{heat}=0.90), host-pathogen (p_{cure}=0.15), and stock market (p_{invest}=0.95) models with 1% training data. The results are given in the table below.
>
> | Model  | Forest Fire | Forest Fire    | Host-Pathogen |  Host-Pathogen   | Stock Market | Stock Market    |
> |---------------------|:----------------------:|:---:|:------------------------:|:---:|:-----------------------:|:---:|
> |                                    |           F1            | AUC                   |            F1            | AUC |            F1           | AUC |
> |Attention-CA            | 0.8305±0.0822 | 0.9705±0.0283 | 0.6759±0.1159 | 0.9433±0.0372 | 0.5103±0.1862 | 0.7692±0.0829 |
> |RA-NCA (RNN)         | 0.8410±0.0791 | 0.9725±0.0266 | 0.6547±0.1199 | 0.9373±0.0376 | 0.5692±0.1857 | 0.7756±0.0848 |
> |RA-NCA (LSTM)       | 0.8672±0.0729 | 0.9768±0.0220 | 0.6957±0.1078 | 0.9443±0.0405 | 0.5762±0.1887 | 0.7751±0.0855 |
>
> Attention+RNN is noted as RA-NCA (RNN). Since we aim to understand the efficacy of the memory module within the RA-NCA architecture, we did not focus on exactly matching parameters of RNN based RA-NCA (12,729) with that of LSTM based RA-NCA (19,065).

---

> ### Author Response · Authors · 2023-11-23
> **Response to the Reviewer o3LN 3/7**
>
> The F1 score and AUC are generally higher in both RA-NCA (RNN and LSTM) models except for the host-pathogen. Our primary observation is that the accuracy gap between Attention-CA and RA-NCA (LSTM) is higher in the forest fire and stock market, indicating the memory function (LSTM) plays an important role in these environments than the host-pathogen. This might be because the forest fire model and stock market involve more complex interactions, such as dynamic heat dissipation and accumulation (forest fire) and dynamic inactive state (stock market). In this light, introducing the memory function will easily improve the accuracy in the forest fire and stock market. However, as RNN is often unstable due to the lack of cell states, this may make the prediction rather unstable in less dynamic systems such as our host-pathogen. In summary, introducing the memory function is generally helpful if the systems involve the complex behaviors. We summarize the discussion and experimental results in Appendix B (RNN-based RA-NCA network) (see Table 7) and also mention "Note, RA-NCA with a basic RNN module (not LSTM) is investigated in Appendix B." in the section 3.1 (Recurrent Attention-based Neural Cellular Automata Network).
>
> > I would have also liked to see a comparison to the obvious state-of-the-art architecture Attention-Transformer NCA (Tesfaldet et al., 2022)
>
> > It would be encouraged to show a direct comparison to the Transformer-NCA
>
> > Is there a reason why the Transformer Architecture is not applicable in this setting? Would you consider using the Attention-Transformer NCA from Tesfaldet et al., 2022 as a more competitive baseline?
>
> **Reply**: We can apply the Transformer architecture into the Cellular Self-Attention module. However, we are already using a multi-head attention module (with a single head) followed by a feed-forward network, which is the basic building block of Transformer. The difference is that skip connection and normalization layers are not considered in our cellular self-attention (see Figure 3(b) right). We should have clarified the details on the implementation. We upload our codebase (model.py) in the supplement to ease the understanding of our model architectures. Clean codebase and manual will be provided after the final decision.
>
> We agree that Attention-Transformer NCA is the state-of-the-art model particularly for their applications, such as image denoising (stationary). However, it may not be a suitable architecture for our applications (dynamic) as discussed below. Let us consider applying Attention-Transformer NCA to our prediction problems by recursively predict the sequential images (i.e., using its output as next input).
>
> The Attention-Transformer NCA aims to update cells towards a stable “fixed-point” (Tesfaldet et al., 2022). For example, to generate a final image, the NCA model receives an input image, returns an updated image, and then utilizes it as the next input image. It repeats this process by 8 ~ 32 steps and generate almost no updates after 32 steps. In image denoising, once a noisy image is fully denoised, the model should not update pixels anymore (otherwise causing a divergence). Hence, ensuring a “fixed-point” is an important goal as it addresses stationary data. However, it poses a challenge in predicting dynamical systems, where the model should generate a “sequence”. In other words, after the model generate an image at ‘t’, it should be able to continually generate the next images at ‘t+1’, ‘t+2’, … , instead of staying at ‘t’ treating it as a fixed-point.
>
> Further, Attention-Transformer NCA will have high computational complexity in both training and inference. It requires 8 ~ 32 steps to generate a single image for training, and 64 steps for inference. For the forest fire model, where the model needs to generate 50 prediction images, the computational graph to calculate its gradient can be very long (i.e., up to 32x50=1600 sequences). It significantly increases the GPU memory and training time as well. Similarly, the model should repeat the update steps up to 64*50=3200 times for inference, which still requires a large GPU memory and takes a long inference time.

---

> ### Author Response · Authors · 2023-11-23
> **Response to the Reviewer o3LN 4/7**
>
> To illustrate the preceding challenges, we adopted Attention-Transformer NCA to the deterministic forest fire model with full training data. Due to the abovementioned computational difficulties, we set the prediction timesteps to 10 (observe 10 frames and predict 10 frames). For the implementation, we employ the codebase attached in the supplement of the paper (https://openreview.net/forum?id=9t24EBSlZOa). We modify the model size (23,731 parameters) to be similar with our RA-NCA by setting the embedding dimension to 64 and MLP dimension to 16. Please see Figure 15 (training dynamics) and Figure 16 (prediction results) in Appendix. Attention-Transformer NCA completely fails to predict the diffusion of fire.
>
> In summary, our RA-NCA follows the common architectural strategy in NCA, operating on the Moore’s neighborhood, but our problem cannot be effectively and efficiently solved by Attention-Transformer NCA models. Also, we would like to mention that the above-mentioned two challenges are applied to the other existing NCA models (non-recurrent), but such discussion has been missed since their applications are focused on the stationary data. We add this discussion and experimental result in the new paragraph of Appendix C (Comparison with other NCA networks - Challenges in Employing prior NCA networks) and also briefly provide the discussion in the section 2 (Related Work and Background - Neural Cellular Automata).
>
> > Apart from the notation and the section on Multi-Agent interaction, I also see no distinction being made between classical neural CAs and each cell being an agent.
>
> **Reply**: Our term “agent” refers to an element with inherent dynamics and local interaction. Along with the above answer, each cell being an agent indicates that our focus lies in the dynamical systems, and it makes the distinction from conventional neural CAs since they are focused on the stationary data without the agent notion.
>
>
> > However, since the CA-video is reduced to minimal color diversity that simply represent very few states (binary even, as it is evaluated / trained), apart from the concept of "predicting batches of pixels from previous batches of pixels", I see not much of a connection to the field of NCAs.
>
> **Reply**: The reviewer’s comment is reasonable. We understand that the prior works on NCAs often generate continuous values (or many states), such as image denoising [1]. However, at the same time, we think that the field of NCAs is evolving with other interesting applications that are driven by only few discrete states. For example, point cloud reconstruction [2] (each point is represented by a binary state: presence or not), damage recovery of soft robots [3] (morphology of robots is implemented by few-state pixels), and game map generation [4] (map is implemented by few-state pixels). In this light, RA-NCA can be connected to this emerging field of NCAs, and thereby our empirical discoveries from the RA-NCA architecture, primarily data efficiency and scalability, can provide important research insight in this field. We discuss the application of NCAs in discrete systems in section 2 (Related Work and Background - Neural Cellular Automata)
>
> [1] Tesfaldet, Mattie, Derek Nowrouzezahrai, and Chris Pal. "Attention-based Neural Cellular Automata." Advances in Neural Information Processing Systems 35 (2022): 8174-8186.
>
> [2] Grattarola, Daniele, Lorenzo Livi, and Cesare Alippi. "Learning graph cellular automata." Advances in Neural Information Processing Systems 34 (2021): 20983-20994.
>
> [3] Horibe, Kazuya, Kathryn Walker, and Sebastian Risi. "Regenerating soft robots through neural cellular automata." Genetic Programming: 24th European Conference, EuroGP 2021, Held as Part of EvoStar 2021, Virtual Event, April 7–9, 2021, Proceedings 24. Springer International Publishing, 2021.
>
> [4] Earle, Sam, et al. "Illuminating diverse neural cellular automata for level generation." Proceedings of the Genetic and Evolutionary Computation Conference. 2022.
>
> > Also, the discussion on original cell-states on page 14 in the Appendix could perhaps deserve a mention in the main paper
>
> **Reply**: Thanks for the suggestion. We add more details on the cell states in the section 2 (Related Work and Background - Dataset: Synthetic Multi-agent System) from the Appendix A (Dataset).

---

> ### Author Response · Authors · 2023-11-23
> **Response to the Reviewer o3LN 5/7**
>
> > I would recommend to shift the focus away from multi-agent interactions and video-prediction networks to focus and elaborate more on the two core-insights of this paper (Low-Data Efficiency and Scalability) to improve focus and balance of the paper. The scaling aspect in particular could offer an interesting insight into the degrees of scalability.
>
> **Reply**: Thank you for the very insightful feedback. We agree with the reviewer’s suggestion. Accordingly, we perform more comprehensive experiments for the scalability. We primarily explored two questions on the scalability: 1) how the prediction accuracy in the 512x512 scale (test) changes if the model is trained in even smaller than the 64x64 scale (train). 2) how the baseline models (ConvLSTM-CA and Attention-CA) in the 64x64 scale (train) perform in the 512x512 scale (test). The following discussion and experimental results are provided in the new paragraph of section 4.1 (Comparison with other NCA Networks - Scalability: Evaluation in Large Stochastic Systems).
>
> For the first task, we compare the models trained in three scales, 32x32, 48x48, and 64x64. The training environments are stochastic forest fire (p_{heat}=0.9), host-pathogen (p_{cure}=0.15), stock market (p_{invest}=0.95) with the full amount of data. The results are given in the table below:
>
> | Model (train->test) | Forest Fire | Forest Fire    | Host-Pathogen |  Host-Pathogen   | Stock Market | Stock Market    |
> |---------------------|:----------------------:|:---:|:------------------------:|:---:|:-----------------------:|:---:|
> |                                   |           F1            | AUC                   |            F1            | AUC |            F1           | AUC |
> | RA-NCA (64->512)    |  0.9172±0.0196 | 0.9878±0.0079  | 0.7241±0.0403 | 0.9593±0.0220 | 0.6177±0.1467 | 0.7758±0.0706  |
> | RA-NCA (48->512)    |  0.9163±0.0187 | 0.9872±0.0085  | 0.7196±0.0427 | 0.9585±0.0225 | 0.6175±0.1467 | 0.7758±0.0705  |
> | RA-NCA (32->512)    |  0.9114±0.0215 | 0.9852±0.0088  | 0.7161±0.0422 | 0.9579±0.0227 | 0.6112±0.1501 | 0.7756±0.0705  |
>
> Interestingly, we observe that the accuracy is not significantly different in the three models. There is improvement by increasing the training scale, but the accuracy disparity between 32->512 and 64->512 cases are marginal. This results provide an important insight in learning the locally-interacting systems, since reducing the training scale from 512 to 32 can significantly save the training computational costs.
>
> We had a question on whether this similar accuracy will be preserved in the data-limited scenarios as well. We perform the same experiment with 1% training data. The results are given below:
>
> | Model (train->test) | Forest Fire | Forest Fire    | Host-Pathogen |  Host-Pathogen   | Stock Market | Stock Market    |
> |---------------------|:----------------------:|:---:|:------------------------:|:---:|:-----------------------:|:---:|
> |                                   |           F1            | AUC                   |            F1            | AUC |            F1           | AUC |
> | RA-NCA (64->512)    |0.8921±0.0194|0.9778±0.0109|0.7061±0.0446|0.9463±0.0275|0.5775±0.1633|0.7728±0.0714|
> | RA-NCA (48->512)    |0.8782±0.0228|0.9728±0.0127|0.6746±0.0531|0.9215±0.0269|0.5422±0.1742|0.7704±0.0729|
> | RA-NCA (32->512)    |0.8851±0.0217|0.9635±0.0132|0.6262±0.0595|0.9239±0.0395|0.5660±0.1617|0.7693±0.0728|
>
> Now, the accuracy disparity across the different training scales is not marginal. More importantly, we would like to point out that the degradation induced by 100%->1% training data is more obvious in the model trained in lower scales. For example, in Host-Pathogen, the F1 score of 32->512 case drops by 0.089 by reducing the training data 100->1%, while that of 64->512 case drops by 0.018. From this result, we view the data efficiency and system scales are correlated. There are more chances for the model to observe the cell transitions in higher scales. For example, 1% training data of 64x64 will involve approximately 4 times more cell transitions than 1% training data of 32x32.

---

> ### Author Response · Authors · 2023-11-23
> **Response to the Reviewer o3LN 6/7**
>
> For the second task, we train ConvLSTM-CA and small Attention-CA in 64x64 scale and evaluate in 512x512 scale across the three stochastic datasets with the limited training data (1%). The results are as follows:
>
> | Model (64->512) | Forest Fire | Forest Fire    | Host-Pathogen |  Host-Pathogen   | Stock Market | Stock Market    |
> |---------------------|:----------------------:|:---:|:------------------------:|:---:|:-----------------------:|:---:|
> |                                   |           F1            | AUC                   |            F1            | AUC |            F1           | AUC |
> | ConvLSTM-CA | 0.8638±0.0225 | 0.9708±0.0161 | 0.5935±0.0672 | 0.8982±0.0511 | 0.4663±0.1825 | 0.7545±0.0759 |
> | Attention-CA | 0.8588±0.0269 | 0.9730±0.0157 | 0.6903±0.0487 | 0.9438±0.0249 | 0.5104±0.1668 | 0.7667±0.0699 |
> | RA-NCA   |0.8921±0.0194|0.9778±0.0109|0.7061±0.0446|0.9463±0.0275|0.5775±0.1633|0.7728±0.0714|
>
> In summary, our RA-NCA is still performing better than the ConvLSTM-CA and Attention-CA in large scale systems. The experimental results are summarized in Table 2 and 3. We also provide the visualization of prediction results in the large systems from the three networks in Figure 5. As the reviewer suggested, we compress the discussion and emphasis on video prediction networks (section 4.2) to shift the focus towards the data efficiency and scalability.
>
> > Why was 32 chosen as the encoding dimension?
>
> **Reply**: We explore other encoding dimensions of RA-NCA. The encoding dimension (n) is associated across the feature extractor (R^{3}->R^{n}), Recurrent cellular attention module (R^{n}->R^{n}), and the decoder (R^{n}->R^{1}) in RA-NCA. We consider three encoding dimensions, 16 (6,185 parameters) , 32 (19,065 parameters), and 48 (39,113 parameters), in stochastic forest fire (p_{heat}=0.90), host-pathogen (p_{cure}=0.15), and stock market (p_{invest}=0.95) models with the 1% amount of training data. The comparison table is given below:
>
> | Model  | Forest Fire | Forest Fire    | Host-Pathogen |  Host-Pathogen   | Stock Market | Stock Market    |
> |---------------------|:----------------------:|:---:|:------------------------:|:---:|:-----------------------:|:---:|
> |                                   |           F1            | AUC                   |            F1            | AUC |            F1           | AUC |
> | RA-NCA (n=16) | 0.8335±0.0934 | 0.9696±0.0292 | 0.5560±0.1909 | 0.7738±0.0852 | 0.6810±0.1160 | 0.9287±0.0479 |
> | RA-NCA (n=32) | 0.8672±0.0729 | 0.9768±0.0220 | 0.5762±0.1887 | 0.7751±0.0855 | 0.6957±0.1078 | 0.9443±0.0405 |
> | RA-NCA (n=48) | 0.8636±0.0729 | 0.9758±0.0224 | 0.5848±0.1821 | 0.7756±0.0852 | 0.7001±0.1087 | 0.9485±0.0372 |
>
> We observe relatively higher accuracy gap between n=16 and n=32 models, than between n=32 and n=48 models. It implies that the limited capacity of n=16 model is relieved from n>= 32 models. From this result, we choose n=32 as a baseline dimension (compact but not too small capacity). This discussion and results are provided in the Appendix B (Various Encoding Dimensions) (see Table 5)
>
> *The n=48 model is worse than the n=32 model in the stochastic forest fire, but better in the host-pathogen and stock market datasets. It might be because 1% training data is too small compared to the larger capacity of n=48 model, particularly in the forest fire dataset. Note, the models predict 50 frames in the forest fire, while predict 20 frames in the host-pathogen and stock market.

---

> > ### Author Response · Authors · 2023-11-23
> > **Response to the Reviewer o3LN 7/7**
> >
> > > Could you please specify on the ⊙ operator in Eq. 4?
> >
> > **Reply**: It represents element-wise multiplication. We add a description below the equation. We apologize for the missing description on the operator.
> >
> > > Minor Comments: p3: “Recently, graph-based, attention- based, variational autoencoder-based NCA networks are also proposed.” could use citations, as could the “classical” ML-LSTM literature (e.g. Schmidthuber et al) that you are building on.
> >
> > **Reply**: Thank you for the comment. We added references for the ML-LSTM literature [1] and other NCA networks [2-4] in the introduction and propose method sections.
> >
> > [1] Hochreiter, Sepp, and Jürgen Schmidhuber. "Long short-term memory." Neural computation 9.8 (1997): 1735-1780.
> >
> > [2] Grattarola, Daniele, Lorenzo Livi, and Cesare Alippi. "Learning graph cellular automata." Advances in Neural Information Processing Systems 34 (2021): 20983-20994.
> >
> > [3] Tesfaldet, Mattie, Derek Nowrouzezahrai, and Chris Pal. "Attention-based Neural Cellular Automata." Advances in Neural Information Processing Systems 35 (2022): 8174-8186.
> >
> > [4] Palm, Rasmus Berg, et al. "Variational neural cellular automata." arXiv preprint arXiv:2201.12360 (2022).
> >
> > > Comparable (or better) performance of the CNN+LSTM in Fig.5 (forest fire) up until data-amount of around 10% could be more discussed, as could the good results on the other two domains (that are only found in Table 5 in the Appendix).
> >
> > **Reply**: Thank you for the suggestion. ConvLSTM-CA shows promising results until 5% of training data, particularly in Gaussian Forest. One possible explanation for Gaussian Forest is that, CNN might capture the non-uniform tree pattern better than the Attention-based models. Also, with the full training data, it achieves better accuracy than the others in Stock Market model. We mention these observations is in the section 4.1 (Data Efficient Deterministic Interaction Learning, Data Efficient Stochastic Interaction Learning).
> >
> > > I would also like to see the std-div. in the main paper tables (similar to the ones in the Appendix) and some more discussion on variance, since e.g. Fig.6 (the zoomed out bar-plots) shows how close performance is if you consider the variance.
> >
> > **Reply**: Thank you for the suggestion. We include the std-div in all the tables in the main paper and Appendix as well. We also mention "RA-NCA generally provides better predictions than the baselines, but in those cases (particularly AUC), there could be some overlaps with the baselines in the accuracy distribution" in the section 4.1 (Data Efficient Stochastic Interaction Learning).

---

> > > ### Comment · Reviewer_o3LN · 2023-12-01
> > >
> > > I thank the authors for their in-depth replies and the work they have put into this revision, I am glad that some of the advice was agreeable and taken to heart. Due to the full extent of these changes I feel, however, that this paper differs so much from the initial draft, that some refinement and a fresh new round of reviews is needed here. The estimation for the initial submission will therefore not change, but I strongly encourage the authors to polish this revised work and try again at the next opportunity.

---

### Official Review · Reviewer_rEW6 · 2023-11-01

**Soundness:** 3 good
**Presentation:** 2 fair
**Contribution:** 3 good
**Rating:** 5
**Confidence:** 3

**Summary:**

The paper proposes a new Recurrent Attention-based Neural Cellular Automata (RA-NCA) in modeling complex systems involving many agents with local, often stochastic interactions.  RA-NCA's innovation lies in a recurrent cellular attention module that combines long short-term memory (LSTM) with cellular self-attention. By evaluating on three simulated multi-agent datasets, RA-NCA shows three good properties which are 1.) Robustness in the presence of stochastic interactions; 2.) Data Efficiency which requires less training data due to the permutation invariance inductive bias; and 3.) Scalability, as it can be trained on small systems and successfully applied to significantly larger systems without the need for re-training, and without a decrease in performance.

**Strengths:**

1. The paper proposes a new model that extends existing NCA methods to capture long-term local and stochastic interactions among agents.
2. The method is technically sound and the writing is in general easy to follow.
3. The experiment results compared with selected baselines show the superior of the proposed method.

**Weaknesses:**

1. My major question is the comparison with existing baselines. I understand the paper targets developing a new NCA method to address long-term local interaction among agents and the stochastic property. However, for the specific task it is dealing with, there are much more methods to compare with in literature on multi-agent dynamical system modeling. Examples include discrete GNN-based methods [1][2], and continuous GNN-based methods [3][4] where the key idea is similar to capture the influence from neighbors and past timestamps to make predictions in the future. Also there are some reinforcement learning literature that can address the same task. I believe at least the authors should have a thorough discussion about these directions.


[1] Alvaro Sanchez-Gonzalez et.al.  Learning to simulate complex physics with graph networks.


[2] Peter W. Battaglia et.al. Interaction Networks for Learning about Objects,Relations and Physics.


[3] Zijie Huang et.al. Learning continuous system dynamics from irregularly-sampled partial observations.


[4] Chengxi Zang et.al. Neural Dynamics on Complex Networks.

**Questions:**

1.  many-agent --> multi-agent?
2. In Figure 4, can you also plot the visualization from baselines as comparisons?

---

> ### Author Response · Authors · 2023-11-23
> **Response to the Reviewer rEW6 1/3**
>
> We are grateful to the reviewer for their constructive feedback about our work. We hope we have addressed their concerns and questions regarding the paper and hope they will reconsider their rating.
>
> > there are much more methods to compare with in literature on multi-agent dynamical system modeling. Examples include discrete GNN-based methods [1][2], and continuous GNN-based methods [3][4] where the key idea is similar to capture the influence from neighbors and past timestamps to make predictions in the future. Also there are some reinforcement learning literature that can address the same task. I believe at least the authors should have a thorough discussion about these directions.
>
> **Reply**: Thank you for providing the interesting references. We read them in detail and agree with that the discussion on these methods should be included in the paper. However, but our problem and objective are quite different with their works.
>
> The primary differences are as follows:
>
> **1) Single State and Multiple States**: They address physical systems, where agent state is not essentially defined. For example, in sand particles [1], bouncing balls [2], and spring systems [3], the agent is just a particle (single state) and its characteristic (e.g., interaction rule) does not change. However, our systems include multiple-state agents. Their states dynamically change, and agent interaction rules are also different depending on the states. This leads to the next difference below.
>
> **2) Different Objective**: Their goal is mostly predicting the trajectory of single-state agents (i.e., particles) [1-3]. In contrast, our goal is to predict the state “transition” of agents. This requires a different problem formulation since the trajectory is considered a regression problem while the state transition is considered a classification problem.
>
> **3) Data efficiency and Scalability**: Our key novelty is in “identifying the benefits of the novel NCA-based architecture, primarily data-efficiency and scalability, in locally interacting multi-agent systems.” While [1] showed the scalability, the references did not explore the data efficiency [1-4] or/and scalability [2-4] of their methods.
>
> **4) Stochastic System**: In addition, their systems are driven by certain physical equations, such as heat diffusion [4]; hence the system evolution is deterministic, and stochastic interaction is not considered. However, this paper aims to explore the various stochasticity levels in the agent interaction.
>
> In terms of technical differences,
>
> **[1]** does not have a temporal encoding module for each of agents. This intuitively makes sense since their agents (particles) do not change the interaction rules. In contrast, our systems involve the interaction rules that depend on the dynamic state of agents. Thus, their method may not be effective to learn the agent-wise dynamics driven by multiple-state agents.
>
> **[2-3]** require prior knowledge on the system, such as the interaction graph that defines which agents are interacting each other. Also, it is not suitable for the large-scale systems since the computational costs quadratically (or higher) increase with the number of agents [5]. However, RA-NCA does not require the interaction graph, and computational costs increase linearly with the number of agents as the operation is constrained on the Moore’s neighborhood.
>
> **[4]** introduces a linear diffusion operator (page 4) which dimension is R^{nxn}. Similarly, this potentially poses a challenge to address a large system. For example, if we evaluate their methods in 512x512 scale (the number of agents (n) is 262,144), the operator dimension becomes R^{262,144x262,144}. That is, the computational costs quadratically increases with the number of agents.

---

> > ### Author Response · Authors · 2023-11-23
> > **Response to the Reviewer rEW6 2/3**
> >
> > There are some related literatures on reinforcement learning (RL), as Reviewer kzFw also suggested. For example, [6] addressed fully-decentralized environments, where each agent is only allowed to access local information and controlled by a local controller. However, these works are generally focused on the control problem, not the prediction problem. The authors in [6] also stated “This framework of networked multi-agent systems finds a broad range of applications in distributed cooperative control problems.” (page 7 [6]). More importantly, RL-based approaches often require large training samples [7]. As our primary goal is to demonstrate the efficacy of RA-NCA in extremely data-limited environments, we mainly study the recurrent neural networks-based predictive modeling in this paper.
> >
> > In summary, we agree that more discussion on other methods (non-NCA based) will improve the quality of the work, and accordingly, we add the discussion on the suggested references in section 2 (Related Work and Background).
> >
> > [1] Alvaro Sanchez-Gonzalez et.al. Learning to simulate complex physics with graph networks.
> >
> > [2] Peter W. Battaglia et.al. Interaction Networks for Learning about Objects,Relations and Physics.
> >
> > [3] Zijie Huang et.al. Learning continuous system dynamics from irregularly-sampled partial observations.
> >
> > [4] Chengxi Zang et.al. Neural Dynamics on Complex Networks.
> >
> > [5] Hoshen, Yedid. "Vain: Attentional multi-agent predictive modeling." Advances in neural information processing systems 30 (2017).
> >
> > [6] Zhang, K., Yang, Z., Liu, H., Zhang, T., & Basar, T. (2018, July). Fully decentralized multi-agent reinforcement learning with networked agents. In International Conference on Machine Learning (pp. 5872-5881). PMLR.
> >
> > [7] Micheli, Vincent, Eloi Alonso, and François Fleuret. "Transformers are sample efficient world models." arXiv preprint arXiv:2209.00588 (2022).

---

> > > ### Author Response · Authors · 2023-11-23
> > > **Response to the Reviewer rEW6 3/3**
> > >
> > > > many-agent --> multi-agent?
> > >
> > > **Reply**: We apologize if the term “many-agent” confused the reviewer. We tried to emphasize the large scale of our systems. We unify the term to “multi-agent” throughout the paper.
> > >
> > > > In Figure 4, can you also plot the visualization from baselines as comparisons?
> > >
> > > **Reply**: Thank you for the suggestion. We change the figure to show the prediction results of all three models (RA-NCA, Attention-CA, and ConvLSTM-CA) across the three datasets now in Figure 5.

---

### Author Response · Authors · 2023-11-23
**Summary of the Revision**

We would like to thank the reviewers for their invaluable assessments and insightful suggestions regarding our paper. We deeply appreciate the constructive feedback provided, which guided us to improve the quality of the paper. The changes in the revised paper are highlighted in blue.

Before summarizing the changes made in the revised version, we wish to clarify the main contribution of this paper. Our key technical novelty lies in “identifying the benefits of the novel NCA-based architecture, primarily data-efficiency and scalability, in locally interacting multi-agent systems.” It provides an important yet underexplored insight to address the multi-agent systems using NCAs. We believe that our work can expand the benefits of NCAs to the applications in dynamical systems.

Based on their suggestions, we have made the following changes to the main paper:
* In introduction, our technical novelty is further clarified.
* In section 2 (Dataset: Synthetic Multi-agent System), the definition of our multi-agent systems and agent interaction are clarified.
* In section 2 (Neural Cellular Automata), discussion on the challenges in the existing NCAs, including the state-of-the-art NCA is added.
* In section 2 (Interaction Learning in Multi-agent System), discussion on other graph-based methods for complex physical systems and reinforcement learning-based approaches are added.
* Figure 4 integrates the previous Figure 5 and 6 to clearly explain the data efficiency in a single plot.
* Figure 5 includes the prediction results on RA-NCA, Attention-CA, and ConvLSTM-CA across the three different large systems.
* In Table 2, added more results to show that RA-NCA trained on even smaller systems (<64x64) can be generalizable to large systems (512x512).
* In section 4.1 (Comparison with other NCA Networks), a paragraph (Scalability: Evaluation in Large Stochastic Systems) is added to discuss the scalability in detail.
* In section 4.2 (Comparison with Video Prediction Networks), the contents are compressed and moved to Appendix D.

The changes made to the appendix include:
* A new Appendix B (Analysis on RA-NCA network) is included to discuss the various encoding dimensions, different neighborhood size, and a basic RNN module (instead of LSTM) in RA-NCA architecture.
* A new paragraph in Appendix C (Challenges in Employing prior NCA networks) is added to discuss the technical bottlenecks to adopt existing NCAs to dynamical systems and explore the state-of-the-art NCA in our problem.
* A new paragraph in Appendix C (Large Attention-CA network) is added to demonstrate the performance of Attention-CA with the more trainable parameters.
* A new Appendix E is included to demonstrate the performance of RA-NCA in the real-world wildfire dataset.